# Photothermal-enabled single-atom catalysts for high-efficiency hydrogen peroxide photosynthesis from natural seawater

Wei Wang[1,2,4], Qun Song [3,4], Qiang Luo [1] ✉, Linqian Li[1], Xiaobing Huo[1], Shipeng Chen[1], Jinyang Li[1], Yunhong Li[1], Se Shi[1], Yihui Yuan[1], Xiwen Du [2], Kai Zhang[3] & Ning Wang [1] ✉

Hydrogen peroxide ($H_2O_2$) is a powerful industrial oxidant and potential carbon-neutral liquid energy carrier. Sunlight-driven synthesis of $H_2O_2$ from the most earth-abundant $O_2$ and seawater is highly desirable. However, the solar-to-chemical efficiency of $H_2O_2$ synthesis in particulate photocatalysis systems is low. Here, we present a cooperative sunlight-driven photothermal-photocatalytic system based on cobalt single-atom supported on sulfur doped graphitic carbon nitride/reduced graphene oxide heterostructure (Co−CN@G) to boost $H_2O_2$ photosynthesis from natural seawater. By virtue of the photothermal effect and synergy between Co single atoms and the heterostructure, Co−CN@G enables a solar-to-chemical efficiency of more than 0.7% under simulated sunlight irradiation. Theoretical calculations verify that the single atoms combined with heterostructure significantly promote the charge separation, facilitate $O_2$ absorption and reduce the energy barriers for $O_2$ reduction and water oxidation, eventually boosting $H_2O_2$ photoproduction. The single-atom photothermal-photocatalytic materials may provide possibility of large-scale $H_2O_2$ production from inexhaustible seawater in a sustainable way.

Hydrogen peroxide ($H_2O_2$) is a powerful and environmentally compatible oxidant in many of the world's most vital industries (e.g., medicine, chemical synthesis, and environmental pollution)[1–4]. Also, $H_2O_2$ is expected to grow in importance as a carbon-neutral high-energy liquid energy carrier (60 wt.% $H_2O_2$ aqueous yields an energy density of about 3.0 MJ L$^{-1}$) in terms of its storability and transportability[5,6]. Traditional anthraquinone process for $H_2O_2$ production needs high energy consumption and yields a lot of chemical waste[7,8]. Particulate photocatalysis, namely the catalytic conversion of $O_2$ and water by irradiating solar energy on semiconductors, has emerged as a sustainable strategy for $H_2O_2$ production[9,10]. Generally, $H_2O_2$ can be photocatalytically produced through a selective two-electron $O_2$ reduction (Eq. (1)) and/or two-electron water oxidation (Eq. (2))[5]. Thermodynamically, the two-electron $O_2$ reduction coupled with the four-electron mediated water oxidation (Eq. (3)) is more favorable for $H_2O_2$ photosynthesis.

$$O_2 + 2e^- + 2H^+ \rightarrow H_2O_2 \ (0.695 \text{ V versus NHE}) \tag{1}$$

$$2H_2O \rightarrow 2H^+ + 2e^- + H_2O_2 \ (1.76 \text{ V versus NHE}) \tag{2}$$

$$2H_2O \rightarrow 4H^+ + 4e^- + O_2 \ (1.23 \text{ V versus NHE}) \tag{3}$$

[1]State Key Laboratory of Marine Resource Utilization in South China Sea, Hainan University, Haikou 570228, P. R. China. [2]Institute of New Energy Materials, School of Materials Science and Engineering, Tianjin University, Tianjin 300072, P. R. China. [3]Sustainable Materials and Chemistry, Department Wood Technology and Wood-Based Composites, University of Göttingen, Göttingen, Germany. [4]These authors contributed equally: Wei Wang, Qun Song. ✉ e-mail: luo-q11@foxmail.com; wangn02@foxmail.com

where NHE represents the normal hydrogen electrode. Unfortunately, the sluggish water oxidation limits overall reaction kinetics of $H_2O_2$ photosynthesis[11,12]. Consequently, additional carbonaceous sacrificial agents (e.g., methanol, isopropyl alcohol, and benzyl alcohol) need be added to match the half-reaction of $O_2$ reduction[13], causing extra difficulty in the product separation. Especially, the pure water feed imposes a heavy burden on the scarce freshwater resource[14,15]. Alternatively, direct $H_2O_2$ photosynthesis from inexhaustible seawater represents a more sustainable development pathway. Recent research on polymer–carbon dot composite and lignin-supported BiOBr photocatalysts showed that $H_2O_2$ photosynthesis from seawater is receiving increasing attention[14,16]. Despite significant advances, such process in real seawater remains limited and is still in its infancy, especially the solar-to-chemical conversion (SCC) efficiency is far from the target for practical application and the stability of photocatalysts suffers from formidable challenges associated with the notorious salt chemistry. It is imperative to develop cost-effective and stable materials to improve the SCC efficiency of $H_2O_2$ photosynthesis from natural seawater.

The steps toward $H_2O_2$ production involve $O_2$ activation and subsequent protonation[17,18]. Thus, the absorption of $O_2$ and the relative energy levels of the reactive intermediates decides whether or how fast $H_2O_2$ photosynthesis will actually happen. It is generally accepted that $H_2O_2$ production could be realized via either a •$O_2^-$-mediated process or a direct $O_2$ reduction pathway[19,20]. Impressively, •$O_2^-$ is typically generated in natural living organisms and can be effectively converted to $H_2O_2$ via redox-active superoxide reductase/dismutase composed of transition metal cofactors and organic ligand moieties[21]. In this regard, heterogeneous single-atom catalysts, containing atomically dispersed metal active sites stabilized on a support with specific geometric and electronic structure, could be potentially considered as the mimics of natural metalloenzyme systems[22,23]. Moreover, the strong metal-support interaction would certainly tailor the electronic structure and induce the charge redistributions[24,25], which could promote the adsorption of $O_2$ and eventually modulate the catalytic reactivity of photocatalysts. On this basis, it can thus be envisioned that, if we could anchor transition metal single atoms on sunlight-driven semiconductors to promote $O_2$ adsorption and activation, the resulting photocatalysts would greatly enhance $H_2O_2$ production. However, incorporating transition-metal single-atom into photocatalytic semiconductors could induce serious carrier recombination due to the half-filled intermediate bands[26,27]. As such, transition metal single-atom photocatalysts need to be rationally designed to ensure the efficient use of photogenerated carriers before the materials become useful for $H_2O_2$ photosynthesis.

Creating heterostructure or hybrid photocatalysts is considered as an essential strategy in the pursuit of advanced photocatalytic processes[28]. Considering the unique structure features and engineerability of two-dimensional (2D) materials, heterostructure photocatalysts based on 2D nanosheets provide considerable opportunities to induce intimate electronic coupling between semiconductor and substrate[29,30], promoting the separation/migration of photogenerated carriers. In addition, broadening photo-response of photocatalysts and ensuring sufficient solar energy utilization are also crucial for improving the SCC efficiency. Notably, interfacial solar heating with a localized photothermal effect by converting low-energy photons into heat offers a promising pathway for incident light usage and becomes necessary for accelerating the reaction kinetics on the surface of adjacent semiconductors[31,32]. Several important chemical reactions were assisted via the corporation of photothermal effect and photocatalysis, such as $CO_2$ conversion[33], hydrogen evolution reaction[34], and nitrogen oxidation[35]. In some cases, the synergistic photothermal-photocatalytic process enhanced the stability of photocatalyst[34]. Thus, a direct integration of photothermal-photocatalytic heterostructure system may provide unprecedented possibilities for efficiently driving

$H_2O_2$ production under sunlight irradiation. So far, such a cooperative system remains less exploited in $H_2O_2$ photosynthesis.

With the above in mind, we propose herein a sunlight-driven single-atom photothermal-photocatalytic system based on atomic-level Co anchored on 2D sulfur doped graphitic carbon nitride/reduced graphene oxide heterostructure (hereafter, referred as Co−CN@G) for non-sacrificial $H_2O_2$ photosynthesis from natural seawater. The single-atom heterostructure can not only suppress the recombination of photogenerated carriers but also endow the photothermal effect to accelerate reaction kinetics. The optimized system achieves an impressive performance in $H_2O_2$ photosynthesis from seawater, affording a high SCC efficiency of 0.72% under simulated sunlight illumination. DFT calculations confirm that atomic-level Co sites together with the intimate interlayer electronic coupling promote the charge separation/migration, $O_2$ reduction and water oxidation, ensuring the greatly enhanced photocatalytic performance in $H_2O_2$ photosynthesis.

## Results

### Photocatalyst synthesis and atomic structure

Co−CN@G nanohybrid was fabricated by in situ thermally polymerizing Co−thiourea complex absorbed on 2D reduced graphene oxide (RGO) sheet (Fig. 1a). The detailed synthetic procedure is given in "Methods". Co−thiourea complex formed by the coordination of cobalt ions with the nitrogen-rich and sulfur-containing thiourea precursor was first absorbed on the structure-directing RGO template. Further heat treatment at an optimized temperature of 400 °C under an argon atmosphere promoted the formation of Co−CN@G nanohybrid. In this way, Co single atom coordinated on sulfur doped $C_3N_4$ (CN) was immobilized on 2D RGO sheet from the polycondensation of Co−thiourea complex, providing sufficient contact area for interfacial electronic coupling. The structure of Co−CN@G was verified by X-ray diffraction (XRD) (Supplementary Fig. 1). A slight shift but stronger peak at 26.7° was observed in contrast with RGO because of the overlap of the (002) peak of CN and RGO, a solid testimony for the strong chemical coupling between Co−CN and RGO[36,37]. No additional diffraction peaks associated with Co were found, whereas the porosity and surface area increased based on the results of $N_2$ sorption analysis (Supplementary Fig. 2). Fourier-transform infrared spectroscopy showed the typical tri-s-triazine, C-N and N-H stretching vibrations of graphitic CN (Supplementary Fig. 3)[38]. Electron microscopy images indicated that Co−CN@G maintained the lamellar morphology without the formation of obvious particles (Supplementary Figs. 4 and 5). Aberration-corrected high-angle annular dark-field scanning transmission electron microscopy (AC-HAADF-STEM) confirmed the atomically dispersed Co single atoms (Fig. 1b). The STEM-coupled energy-dispersive spectroscopy element mapping revealed the homogeneous dispersion of atomic Co, C, N, and S over the whole sample (Fig. 1c). The sample gave a C/N atomic ratio of 2.75, as depicted in Supplementary Fig. 6. The high-resolution Co $2p$ XPS peaks located at 780.9 and 796.3 eV could be assigned to Co $2p_{3/2}$ and Co $2p_{1/2}$, respectively (Supplementary Fig. 7)[39]. C $1s$ spectra in Co−CN@G hybrid displayed an extra peak at 287.3 eV assigned to the out-of-plane orientation of $sp^3$ C-N species (Fig. 1d), manifesting the interfacial chemical interconnectivity between Co−CN and RGO parallel layers[40].

The atomic structure of Co−CN@G was examined. X-ray absorption near-edge structure spectra (XANES) of Co K-edge demonstrated the location of the absorption edge and the intensity of fingerprint peak (7730 eV) for Co−CN@G at the middle of the reference Co foil and $Co_3O_4$ samples (Fig. 1e). These suggested the presence of single Co atoms in oxidized state[41,42]. Extended X-ray absorption fine structure (EXAFS) of Co−CN@G exhibited a dominant peak at ~1.82 Å, which can be separated into two independent peaks assigned to Co−N and Co−S scattering. No metallic Co−Co peak at 2.1 Å could be seen, suggesting the existence of all Co as isolated atoms (Fig. 1f). The FT-EXAFS profiles

(Fig. 1g) and the corresponding fitting results (Supplementary Fig. 9 and Supplementary Table 1) verified that $CoN_1S_3$ (one Co–N bond and three Co–S bonds) should be the dominating structure in Co–CN@G. Also, Co–CN@G displayed a strong wavelet transform maximum focused at $5.1\,\text{Å}^{-1}$ assigned to Co–N/S contribution, which was sharply distinguished from the corresponding maxima of Co foil (Co–Co, $7.1\,\text{Å}^{-1}$) and $Co_3O_4$ (Co–O, $6.2\,\text{Å}^{-1}$) (Supplementary Fig. 10). [15]N solid-state nuclear magnetic resonance (NMR) measurement was employed to acquire the detailed information of N species at the atomic level[43,44]. The [15]N NMR spectra of Co–CN@G exhibited a poor signal-to-noise ratio (Supplementary Fig. 11) due to the interference of conductive RGO and magnetic Co atoms. Alternatively, the pristine Co–CN was evaluated (Fig. 1h). Four typical resonance peaks at -135, -156, -107, and -191 ppm were observed, which could be assigned to the bridged NH, central $NC_3$, $NH_2$, and $NC_2$ in the tri-s-triazine ring, respectively[25,45]. Anchoring Co onto the CN support resulted in a decreased intensity of the 191-ppm NMR peak, suggesting that Co atoms bonded to the $NC_2$ sites on the CN surface.

### Photoelectronic(thermal) property and electronic structure

Ultraviolet–visible diffuse reflectance spectroscopy indicated that Co–CN@G could cover the UV–visible–NIR light region (Fig. 2a). Compared with other samples, Co–CN@G showed an enhanced absorbance, implying that CN@G combined with Co single atom could utilize solar energy more effectively. Moreover, Co–CN@G exhibited a slight redshift in the optical absorption edge, corresponding to the distinct band gap shrinkage according to Kubelka–Munk function (Supplementary Fig. 12). As revealed by valence-band XPS spectra (Supplementary Fig. 13), the conduction band and valence band energy of Co–CN@G were ca. −0.62 and 2.0 eV (versus NHE), respectively. Notably, the narrowed bandgap and favorable energy band levels were enough for driving $O_2$ reduction and water oxidation thermodynamically. The emission peak centered at 436 nm in photoluminescence emission spectroscopy was markedly reduced on Co–CN@G despite its stronger photon absorption (Supplementary Fig. 14), implying that the radiative recombination was greatly suppressed[46]. The time-dependent photo-response curves at open-circuit voltage showed an elevated photocurrent than that of the other catalysts once exposed to the simulated sunlight irradiation (Fig. 2b), revealing that Co–CN@G had superior charge generation and separation capacity. The increased electrolyte temperature induced by photothermal effect could contribute to the gradual increase of photocurrent[47]. The facilitated charge carrier migration could be further confirmed by the decreased electrochemical impedance (Supplementary Fig. 15). We then explored the photothermal property of the samples by monitoring the time-dependent temperature variation of the suspensions under different photoirradiation (Supplementary Fig. 16). Upon continuous simulated sunlight illumination (AM 1.5 G, one sun) for 60 min, the system temperature spontaneously increased from 27.1 to 37.3 °C for CN@G and 38.4 °C for Co–CN@G, evidently higher than the resulting temperature of the blank solution or Co–CN suspension. By equipping with a 720 nm short-pass filter (NIR light was removed, labeled as UV–vis) or a long-pass filter (UV–vis light was removed, labeled as NIR), Co–CN@G suspension under UV–vis and NIR light exposure yielded a decreased temperature of 7.3 and 2 °C respectively, verifying that the photothermal effect mainly originated from NIR light.

We then performed density functional theory (DFT) calculations to understand the interlayer electron transfer and electronic structure of the photocatalysts. The Mulliken charge difference between the adjacent layers in CN@G (-0.041 e⁻) was larger than that in Co–CN (roughly 0.028 e⁻), implying the strong charge transfer between the interlayers in CN@G (Supplementary Fig. 17)[5]. Obviously, the presence of Co-single-atom sites in CN@G drastically rearranged the charge distribution between the neighboring planes of Co–CN@G (Fig. 2c, d), resulting in an electron depletion in the fourth layer and a distinct electron accumulation on the first (0.021 e⁻) and third layer (0.099 e⁻). As a result, Co–CN@G exhibited a significantly larger charge difference between the adjacent layers (-0.124 e⁻). This result meant that the interlayer electronic coupling of Co–CN@G had been notably promoted by incorporating Co single atoms, facilitating charge separation and transfer. The electronic structures were further analyzed using density of states (DOS) (Fig. 2e and Supplementary Fig. 18). The calculated d-band center of Co–CN@G (−0.97 eV) was much higher than

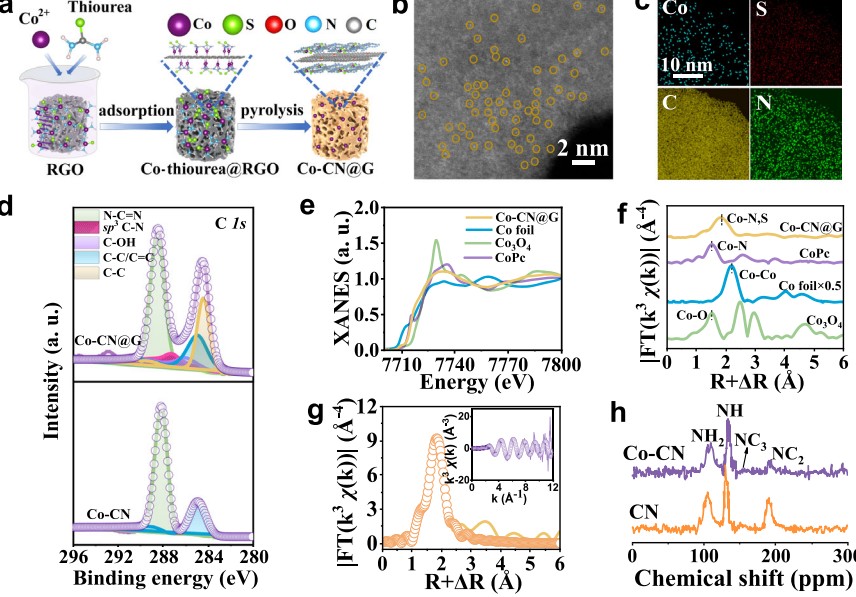

**Fig. 1 | Preparation, characterizations, and analysis of Co–CN@G. a** Schematic illustration of the synthesis of Co–CN@G. **b** HADDF-STEM image. The bright spots of Co atoms are marked with yellow circles. **c** EDS mapping of Co–CN@G. **d** High-resolution C*Is* XPS spectra of Co–CN@G and Co–CN. **e** XANES spectra at the Co K-edge of Co–CN@G, Co foil, $Co_3O_4$, and CoPc. **f** FT $k^3$-weighted $\chi$(k)-function of the EXAFS spectra at Co K-edge. **g** The EXAFS fitting curve for Co K-edge at R space. The inset shows EXAFS fitting curve at k space for Co–CN@G. **h** [15]N solid-state NMR MAS spectra of Co–CN and pristine CN. Source data are provided as a Source Data file.

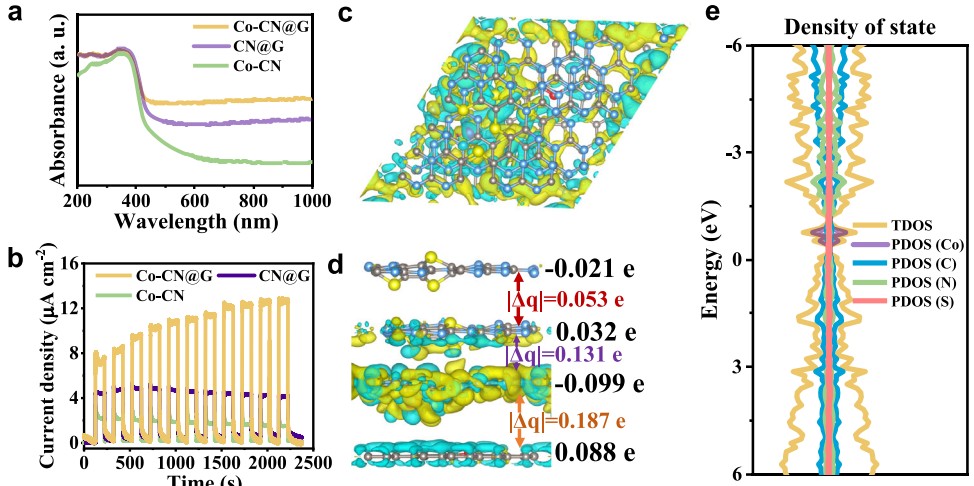

**Fig. 2 | Photoelectronic property and electronic structure of Co−CN@G.**
**a** Diffuse reflectance UV-vis spectra of Co−CN@G, CN@G, and Co−CN.
**b** Transient photocurrent response curves of Co−CN@G, CN@G, and Co−CN
electrodes. The Mulliken charge difference between each adjacent layer of
Co−CN@G for the **c** enlarged top view and **d** cross view. Yellow and cyan iso-

surface represents electron accumulation and electron depletion, respectively.
Blue, gray, yellow, rose, and the red color represents nitrogen, carbon, sulfur,
cobalt, and oxygen, respectively. **e** The calculated total density of states (TDOS)
and partial density of states (PDOS) of Co−CN@G. Source data are provided as a
Source Data file.

that of Co−CN (−1.15 eV), indicating that Co−CN@G was more favor-able to the binding and activation of $O_2$ molecule. Furthermore, the shape and height of peaks in the negative side changed because of the modification of Co $3d$ orbit. Atomically dispersed Co generated two distinct and occupied mid-gap states (-0.5 and 0.7 eV above the valence band maximum), decreasing the band gaps and promoting the localization of photoexcited holes[48]. Additionally, it was found from PDOS that the $3d_{xy}$ orbital of Co atom strongly coupled with the $2p_y/2p_z$ orbital of the N and S atom (Supplementary Fig. 19), further con-firming the strong coordination between Co, N, and S and the forma-tion of in-plane bonds[49].

### Photocatalytic activity for $H_2O_2$ production

Photocatalytic $H_2O_2$ production on different catalysts was first eval-uated in natural seawater under simulated solar irradiation (AM 1.5 G, 100 mW cm$^{-2}$) without sacrificial reagents (Fig. 3a). No $H_2O_2$ was detected in the dark or without photocatalyst, which verified the photo-induced reaction. Under light illumination, Co−CN@G hetero-structure showed a significantly higher $H_2O_2$ yield than Co−CN and CN@G, indicating that Co single atoms combined with CN@G con-tributed to the enhanced photocatalytic reactivity. By optimizing Co loading and mass ratio of components, Co−CN@G with 1.5 wt% Co and a CN/RGO mass ratio of 1:1 produced the largest amount of $H_2O_2$ (Supplementary Fig. 20). Excessive amounts of either Co, Co−CN, or RGO resulted in lower photocatalytic activity, which could be attrib-uted to the formation of Co nanoclusters at high Co loading (Supple-mentary Fig. 21) or insufficient interfacial coupling between Co−CN and RGO. $H_2O_2$ production on the optimized Co−CN@G dramatically increased with time at the early stage of irradiation (Fig. 3b). Notably, Co−CN@G generated 0.44 times higher $H_2O_2$ amount in seawater (8.29 mM) than in pure water (5.77 mM) for the first 10 h. Also, Co−CN@G displayed a lower decomposition rate ($K_d$) of $H_2O_2$ than other samples, and the temperature exerted a negligible effect on the decomposition behavior (Supplementary Fig. 22).

Photosynthesis was further evaluated under different light irra-diation to clarify the effect of incident light sources on $H_2O_2$ produc-tion (Fig. 3c). $H_2O_2$ evolution under NIR light exposure was negligible, indicating that NIR light alone cannot drive $H_2O_2$ formation. Mean-while, irradiating Co−CN@G by the simulated sunlight resulted in a 0.22-fold higher $H_2O_2$ amount than that under UV−vis light

illumination. In other words, NIR light irradiation played a positive effect on the photocatalytic reaction. In addition, the thermal-assisted photosynthesis under UV−vis light illumination was further examined. The $H_2O_2$ amount increased as the reaction temperature elevated, revealing that the thermal effect progressively boosted the photo-catalytic performance of Co−CN@G. Typically, $H_2O_2$ evolved under UV-Vis light illumination at 42 °C was just comparable to the amount under simulated sunlight irradiation, confirming that the localized surface heating through photothermal effect can promote the surface reaction kinetics. To further verify this, photosynthesis experiments were conducted by illuminating two different light sources simulta-neously. A solar simulator (lamp 1) was placed on the one side of the vessel to offer the simulated sunlight illumination (100 mW cm$^{-2}$). The lamp 2 with NIR light export was placed on the other side of the vessel. In this case, the effect of NIR light on the photocatalytic process can be evaluated by adjusting the power of lamp 2 (0, 50, 100 mW cm$^{-2}$). As shown, Co−CN@G produced a higher level of $H_2O_2$ with the increased NIR light intensities (Fig. 3d), while $H_2O_2$ evolved on Co−CN was almost independent of the intensity of NIR light. Therefore, NIR light illumi-nation exerted a vital but auxiliary role in strengthening the catalytic reactivity of Co−CN@G.

The SCC efficiency was determined to be 0.72% under simulated sunlight irradiation (Fig. 3e), which was about 19.8 and 2.5 times higher than that of pristine Co−CN and CN@G (Supplementary Fig. 23), respectively, and superior to other particulate photocatalysts in pure water and natural seawater (Supplementary Table 2). However, the activity definitions used to evaluate catalysts for $H_2O_2$ evolution are not standardized, imposing difficulties in comparing the intrinsic activity of different photocatalysts. The wavelength-dependent apparent quantum efficiency (AQE) of $H_2O_2$ production was then evaluated under the monochromatic light illumination (Fig. 3f). Co−CN@G exhibited a maximum AQE of 9.1% at 420 nm. The photo-catalytic stability was evaluated by collecting and reusing the photo-catalyst for 15 cycles. Co−CN@G was stable enough to maintain its photocatalytic activity during repeated cycles (Fig. 3g). In contrast, the reference samples, especially Co−CN, yielded the gradually decreased $H_2O_2$ amount after continuous cycling (Supplementary Fig. 24). The unchanged XRD pattern, Co $2p$ XPS spectra, FT-IR spectrum and inapparent morphology transformation further confirmed the robust nature of Co−CN@G (Supplementary Fig. 25). The Co content in the

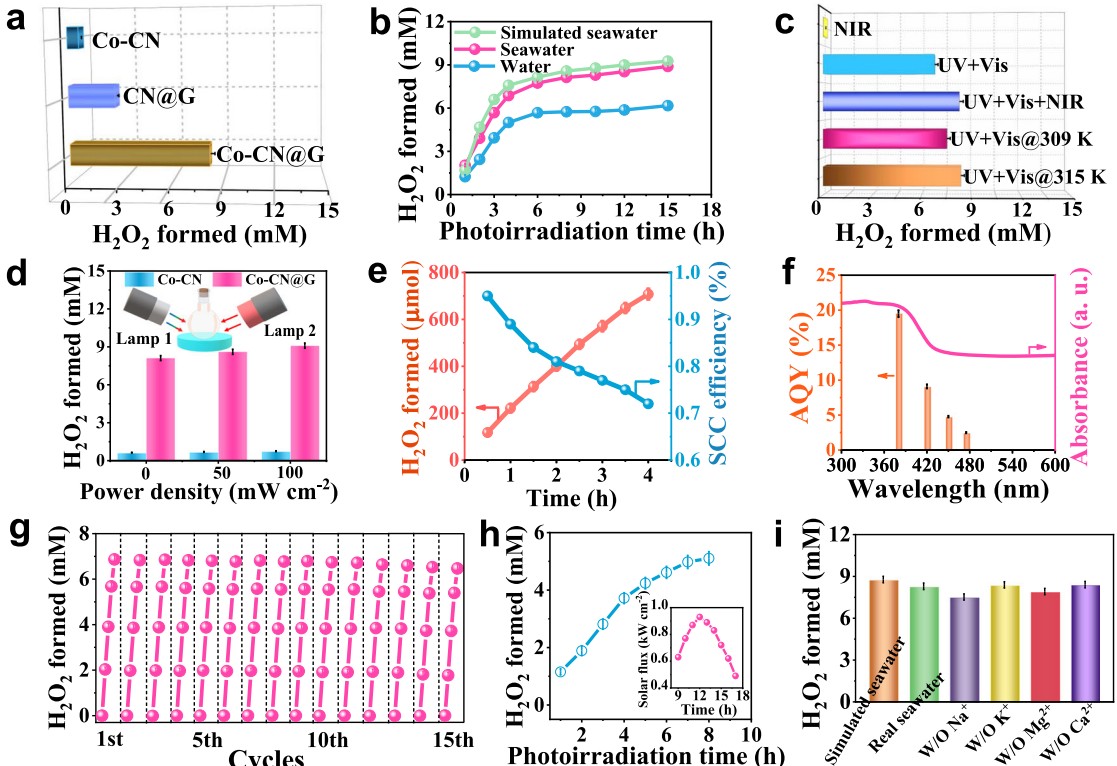

**Fig. 3 | Photocatalytic properties of Co–CN@G. a** Amounts of $H_2O_2$ generated by 10 h of photoirradiation under simulated sunlight (AM 1.5 G, 100 mW cm$^{-2}$) on different photocatalysts in natural seawater. **b** Time course of $H_2O_2$ photo-production by Co–CN@G. **c** Amounts of $H_2O_2$ generated by 10 h of photoirradiation on Co–CN@G under different photoirradiation in natural seawater. **d** Photocatalytic $H_2O_2$ yield on Co–CN and Co–CN@G illuminated by two lamps with adjusting power density. **e** Changes in the amounts of $H_2O_2$ generated on Co–CN@G and the solar-to-chemical (SCC) efficiency under AM1.5 G simulated

sunlight irradiation. **f** Action spectra of Co–CN@G towards $H_2O_2$ production in seawater. **g** Results for a repeated photoreaction sequence with Co–CN@G for 15 cycles. **h** Amounts of $H_2O_2$ generated as a function of time under ambient sunlight irradiation. The inset: Solar flux on 24 July 2022, in Haikou City, China. **i** $H_2O_2$ synthesis from simulated seawater after 10 h in the absence of (W/O) different metal cations. Error bars indicate the standard deviation for three measurements. Source data are provided as a Source Data file.

solution after continuous photocatalytic reaction for 120 h was below the detection limitation of inductively couple plasma mass spectrometer (ICP-MS) measurement, representing no Co leaching. We also performed the outdoor photocatalytic test on 24 July 2022, with an ambient temperature of 28–34 °C and a sunlight intensity of 0.49–0.93 kW m$^{-2}$ in the daytime in Haikou City of Hainan Province, China. Co–CN@G (30 mg in 50 mL seawater) achieved a daily total amount of $H_2O_2$ of 5.1 mM (8508 μmol g$^{-1}$ day$^{-1}$) in the $O_2$-saturated particulate suspension system under ambient sunlight irradiation, demonstrating the potential of large-scale $H_2O_2$ production from seawater (Fig. 3h).

A series of experiments have been performed to elucidate the origin of performance enhancement in seawater. We subjected individual inorganic salt, NaCl, CaCl$_2$, MgCl$_2$, KCl, Na$_2$SO$_4$, and Na$_2$CO$_3$, to the freshwater (Supplementary Fig. 26). It was observed that the presence of these inorganic salts improved $H_2O_2$ production. It has been reported that the electron sink of oxygen-containing groups of carbon materials in the presence of metal cations could boost photocatalytic activity by attracting electrons[14]. To explore the synthetic effect of seawater ions, simulated seawater with 3.4% salinity was used for further investigation since its composition is very close to those of natural seawater. As expected, the simulated seawater showed a similar $H_2O_2$ formation with natural seawater (Fig. 3b). We observed a volcanic relationship between $H_2O_2$ evolution and salinity with a summit at a moderate salinity of 2.4% (Supplementary Fig. 27), illustrating that $H_2O_2$ photosynthesis from natural seawater can be significantly enhanced by pretreatment of mild desalination. The photocatalytic performance was then examined when one cation was removed. To

keep the same ion concentration, ammonium chloride or sulfate was introduced (Supplementary Table 3). We found that the photocatalytic activity was suppressed in the absence of Na$^+$ and Mg$^{2+}$, while the removal of K$^+$ and Ca$^{2+}$ had little effect on $H_2O_2$ formation because their actual concentrations in seawater are low (K$^+$: 0.009 M, Ca$^{2+}$: 0.01 M) (Fig. 3i). DFT calculations were performed to understand the effect of cation on the charge transfer (Supplementary Fig. 28). Hydrated cations were considered to mimic the experimental environment during the calculations, with the optimized number of water molecules bound to Na$^+$ and Mg$^{2+}$ being three and four, respectively. Interestingly, the interlayer charge transfer of Co–CN@G was improved after interacting with Na$^+$(H$_2$O)$_3$ or Mg$^{2+}$(H$_2$O)$_4$ cation. This trend was also confirmed experimentally by resorting time-resolved PL spectroscopy (Supplementary Fig. 29). Co–CN@G treated with different metal chlorides exhibited the increased intensity-average PL lifetime ($\tau$), indicating that the presence of metal cations can improve the efficiency of photoexcited carriers. Next, we examined the reaction when halide ions were absent from simulated seawater (Supplementary Fig. 30). The $H_2O_2$ evolution decreased due to the absence of Cl$^-$ in simulated seawater, indicating that Cl$^-$ exerted a positive effect on $H_2O_2$ generation. The effect of low-concentration Br$^-$ (0.00087 M) can be neglected. A similar enhancement was also observed in a NaCl or NaBr solution. Although Cl$^-$ oxidation by photogenerated holes requires a higher potential than four-electron water oxidation, it is kinetically favorable owing to its two-electron process. The formed Cl$_2$ can further react with water to produce HClO, which is subsequently decomposed into Cl$^-$ and $O_2$ under light irradiation. During the photocatalytic process, Cl$_2$ was not monitored by gas chromatography.

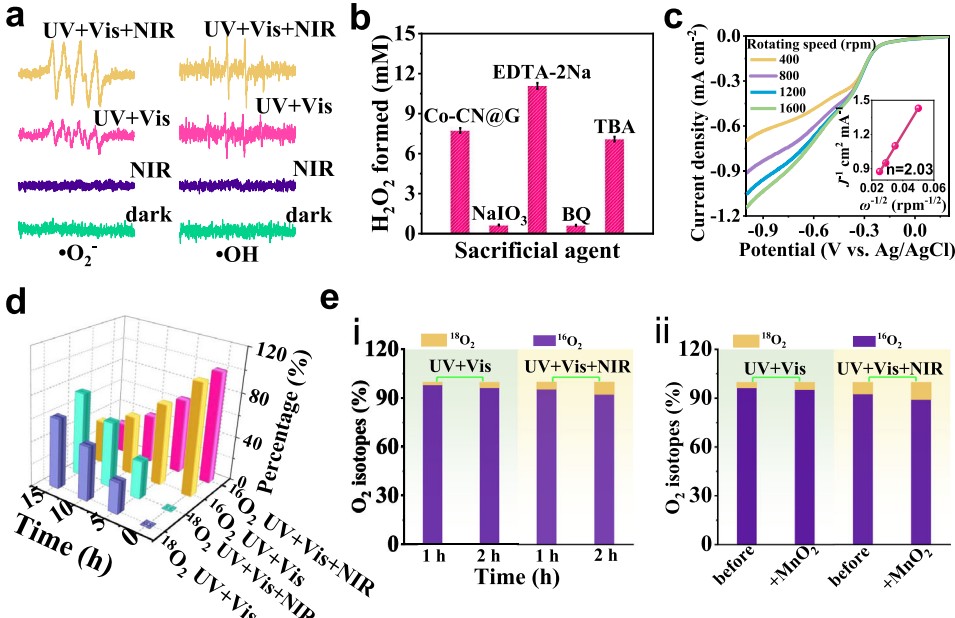

**Fig. 4 | Photocatalytic mechanism of Co−CN@G for H₂O₂ production. a** EPR signals of •O₂⁻ and •OH over Co−CN@G in the presence of DMPO under different photoirradiation. **b** Photocatalytic H₂O₂ generation for 6 h over Co−CN@G with different sacrificial agents. **c** LSV curves of Co−CN@G measured on an RDE at different rotating speeds. Inset: the corresponding Koutecky–Levich plots. **d** ¹⁸O₂ isotope labeling experiments. **e** H₂¹⁸O isotope labeling experiments: **i** the relative intensities of formed ¹⁶O₂ and ¹⁸O₂ after photoirradiation measured by GC-MS; **ii** the relative intensities of ¹⁶O₂ and ¹⁸O₂ in the gas products decomposed from H₂O₂ generated using Co−CN@G. Error bars indicate the standard deviation for three measurements. Source data are provided as a Source Data file.

The net result is that Cl⁻ assists in the four-electron water oxidation. The O₂ amount generated in N₂-saturated NaIO₃ aqueous solution with NaCl additive after 2 h (3.82 μmol) was 0.3 times higher than that with NaF additive (Supplementary Fig. 31), verifying that Cl⁻ has an auxiliary effect on the water oxidation.

## The photocatalytic mechanism for H₂O₂ production

To understand the origin of H₂O₂ evolution over Co−CN@G, a series of control experiments were performed. The generated intermediates during H₂O₂ synthesis were explored using the electron paramagnetic resonance method with 5,5-diemthyl-1-pyrroline *N*-oxide (DMPO) as the spin-trapping agent. No DMPO-•O₂⁻ or DMPO−•OH signal was detected in the dark or under NIR light exposure (Fig. 4a). The signals of DMPO-•O₂⁻ and DMPO−•OH were obviously intense under UV−vis light and simulated sunlight irradiation, and the intensities under simulated solar illumination were higher than those under UV−vis illumination, revealing that the local heating effect generated by photothermal transduction of Co−CN@G motived the generation of radical intermediates. The addition of either NaIO₃ electron acceptor or benzoquinone (BQ) (•O₂⁻ scavenger) into the reaction system resulted in the sharply decreased H₂O₂, meaning that H₂O₂ photosynthesis was dominated by O₂ reduction (Fig. 4b). Rotating disk electrode analysis conducted in an O₂-saturated phosphate buffer solution (0.1 M, pH ~7) indicated that the electron transfer number for O₂ reduction was ~2 (Fig. 4c). Isotopic labeling experiments were further conducted. Co−CN@G photocatalyst in H₂¹⁶O and ¹⁸O₂ mixture was illuminated for different times, and the resulting solution was then Co−incubated with MnO₂ to decompose H₂O₂ into O₂[50]. The evolved gaseous product detected by gas chromatography−mass spectrometry (GC−MS) yielded a strong peak assigned to ¹⁸O₂, and the peak intensity increased with the extension of photoirradiation (Fig. 4d). Moreover, a stronger ¹⁸O₂ signal was observed under simulated sunlight irradiation, illustrating the photothermal-enhanced ¹⁸O₂ reduction over Co−CN@G. Triphenylphosphine (PPh₃) was also employed as a capping agent to examine the oxygen source of H₂O₂ (Supplementary Fig. 32). The intensified ¹⁸O = PPh₃ with respect to ¹⁶O = PPh₃ from

the corresponding m/z result proved that H₂¹⁸O₂ was mainly produced by ¹⁸O₂ reduction. These results testified that H₂O₂ photosynthesis was dominated by two-electron O₂ reduction, and the photothermal effect accelerated the reaction kinetics.

On the other hand, the presence of edetate disodium (EDTA-2Na) in the reaction system exerted a significantly positive effect on the H₂O₂ amount, which could be attributed to the suppressed charge recombination by the EDTA-2Na hole acceptor. Moreover, a significant amount of O₂ was monitored after light-irradiating the N₂-saturated seawater containing NaIO₃ additive, implying that O₂ was exclusively generated from four-electron H₂O oxidation (Supplementary Fig. 33). Interestingly, a small amount of H₂O₂ was also detected in the N₂-saturated seawater (Supplementary Fig. 34), and *tert*-butyl alcohol (TBA) additive (•OH scavenger) showed slightly negative influence on H₂O₂ amount, verifying that the partial H₂O₂ can be produced via the stepwise one-electron water oxidation pathway with •OH as the reactive intermediate. Isotope experiments also exhibited that ¹⁸O₂ was detected when H₂¹⁸O was used after 2 h photoirradiation (Fig. 4e), and increased content of ¹⁸O₂ was also traced by adding MnO₂ into the photoproduced H₂O₂ solution, further implying that H₂¹⁸O was oxidized into both ¹⁸O₂ and H₂¹⁸O₂ over Co−CN@G. Furthermore, the further increase of ¹⁸O₂ under simulated sunlight illumination also indicated that the photothermal effect could accelerate the reaction kinetics of both two-electron and four-electron water oxidation. We then performed DFT calculations to get a further fundamental understanding of the photocatalytic activity of Co−CN@G. As for H₂O₂ production from O₂ reduction, the O₂ chemisorption should be a prerequisite step for its subsequent reduction. Based on the Mulliken population analysis, Co single atoms on CN@G hybrid created polarized active sites on which O₂ adsorption preferably occurred. Under optimized conditions, the negative adsorption energy of O₂ (−0.28 eV) on Co−CN@G was estimated (an exothermic process) (Fig. 5a), which was significantly lower than that on Co−CN ($E_{ads}$ = 1.15 eV) and CN@G ($E_{ads}$ = 1.14 eV) (Supplementary Fig. 35). The coordinative Co−O bond on Co−CN@G with a shorter bond length of 1.95 Å ($d_{Co−O}$ = 2.11 Å in Co−CN) stretched the O = O bond of free O₂ molecule from 1.23 Å to

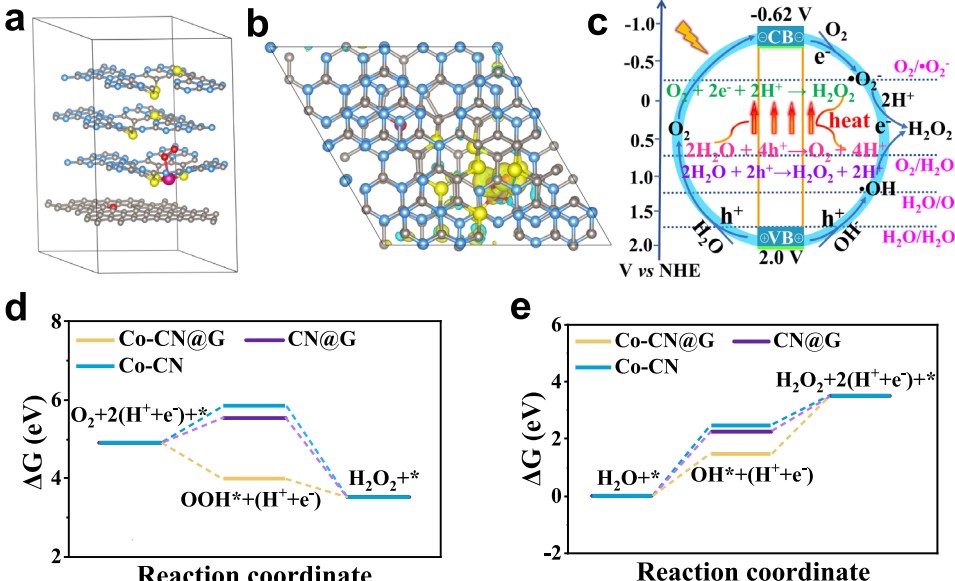

**Fig. 5 | DFT calculations. a** Cross view for $O_2$ adsorption configuration over Co−CN@G. **b** Top view of the charge difference density of $O_2$ adsorption on Co−CN@G. Yellow and cyan iso-surface represents electron accumulation and electron depletion, respectively. Blue, gray, yellow, rose, and the red color represents nitrogen, carbon, sulfur, cobalt, and oxygen, respectively. **c** Mechanism of photothermal−photocatalytic $H_2O_2$ production. The free energy diagram for $H_2O_2$ formation via **d** $O_2$ reduction pathway and **e** water oxidization route on Co−CN@G, CN@G, and Co−CN. Source data are provided as a Source Data file.

1.30 Å. Such bond configurations could enhance the interaction between $O_2$ and multilevel melon plane tremendously[51,52]. Furthermore, notable electron accumulation on the coordinative $O_2$ and electron depletion on the Co sites were found from the charge density difference (Fig. 5b), confirming the occurrence of electron back-donation from Co−single-atom to the absorbed $O_2$. Thus, the photo-generated electrons on the conjugated melon units of Co−CN@G can be donated to the anti-bonding orbital of $O_2$ through the strong electron coupling effect between $O_2$ and Co, boosting $O_2$ activation and subsequent protonation[17]. As shown in Fig. 5c, the photoexcited electrons on the conduction band (CB) can easily drive the step-wise two-electron $O_2$ reduction with •$O_2^-$ as the intermediate. Furthermore, $O_2$ reduction towards $H_2O_2$ formation was also energetically favorable over Co−CN@G (the Gibbs free energy from [$O_2$ + *] to OOH* was only −0.93 eV, a thermodynamic exothermic step) (Fig. 5d). In sharp con-trast, the calculated energy barrier for the intermediate OOH* forma-tion on CN@G and Co−CN was increased by 1.57 and 1.87 eV, respectively. These results suggested that Co−CN@G greatly pro-moted the two-electron $O_2$ reduction process, ensuring efficient $H_2O_2$ photosynthesis. In addition, $H_2O_2$ production from direct water oxi-dation occurred at the N atoms ($NC_2$ site) of the melem unit adjacent to the Co site was also examined (Fig. 5e and Supplementary Fig. 36). The generation of the intermediate state OH* was the most critical step for water oxidation[53]. The results revealed that Co−CN@G sample exhib-ited the smallest $\Delta G$ for the formation of OH* compared to CN@G and Co−CN. Thus, Co−CN@G was also more active for photocatalytic $H_2O_2$ production via water oxidation. Indeed, the electrochemical linear sweep voltammogram (LSV) also demonstrated a higher current for water oxidation on Co−CN@G than on other catalysts (Supplementary Fig. 37). These theoretical calculations implied that the marriage between Co single atoms and CN@G heterostructure favored $H_2O_2$ production from water oxidation and $O_2$ reduction, in consistent with the experimental results.

## Discussion

In summary, we reported a photothermal−photocatalytic Co−CN@G particulate photocatalyst for highly efficient non-sacrificial $H_2O_2$ pro-duction from natural seawater. By taking advantage of the cooperation of CN@G heterostructure and isolated Co atoms, this system exhibited excellent catalytic activity towards $H_2O_2$ photosynthesis. Experimental results also confirmed that the photothermal effect of the Co−CN@G played a positive role in strengthening the photocatalytic activity. A high SCC efficiency of 0.72% was achieved under simulated sunlight irradiation, superior to all currently reported particulate semi-conductors for $H_2O_2$ photosynthesis from pure water and natural seawater. Impressively, when irradiated under outdoor sunlight, the formation rate of $H_2O_2$ reached 8508 µmol g$^{-1}$ day$^{-1}$ in the summer. Theoretical calculations revealed that the synergy between Co single atoms and CN@G heterostructure could promote the interfacial electronic coupling and lower the associated energy barriers of $O_2$ reduction and water oxidation, enabling Co−CN@G to exhibit excel-lent photocatalytic performance for $H_2O_2$ synthesis. This work pro-vides significant guidelines for the rational design of cost-effective photocatalytic systems for future large-scale $H_2O_2$ photosynthesis.

## Methods

### Synthesis of photocatalysts

Reduced graphene oxide (RGO) foam was prepared from GO suspen-sion (0.9 mg mL$^{-1}$, 80 mL) through a hydrothermal method in a Teflon-lined stainless-steel autoclave (100 mL) at 120 °C for 5 h. The resulting foam hydrogel was washed with deionized water and obtained by freeze-drying. Subsequently, the as-prepared RGO foam was soaked into an aqueous solution (30 mL) containing 10 mg of $CoCl_2·6H_2O$ and 500 mg of thiourea for 12 h to facilitate the absorption of Co−thiourea complex on RGO. Afterward, the foam was freeze-dried for 48 h and was treated at 400 °C (ramp rate: 2 °C min$^{-1}$) for 2 h under an Ar atmosphere.

### Photocatalytic reaction

Totally, 50 mg of the photocatalyst was ultrasonically dispersed into the $O_2$-saturated seawater (50 mL) in a sealed jacketed glass reactor. Natural seawater collected from Lingshui Bay (Lingshui City, Hainan, China) was filtered using a 0.22 µm PTFE syringe filter prior to use. The photocatalytic reaction was conducted by photo-irradiating the stir-ring solution with 3A solar simulator equipped with different light filters. To evaluate the stability of photocatalysts, the solid samples

were recovered by centrifugation and re-dispersed in the $O_2$-saturated seawater for the next cycle. The simulated seawater was obtained by adding various inorganic salts (NaCl: 27.21 g; $MgCl_2$: 3.81 g; $MgSO_4$:1.66 g; $CaSO_4$: 1.40 g; $K_2SO_4$: 0.58 g; $K_2CO_3$: 0.21 g; $MgBr_2$: 0.08 g) into 1 L of ultrapure water. The formed amount of $H_2O_2$ was determined according to the colorimetric N, N-diethyl-phenylenediamine (DPD) method[54]. The wavelength-dependent apparent quantum efficiency (AQE) was measured under monochromatic light illumination, with AQE determined using Eq. (4)[55]. The SCC efficiency ($\eta$%) was evaluated under AM 1.5 G light illumination according to Eq. (5)[56]:

$$AQE(\%) = \frac{\text{formed amount of } H_2O_2 \times 2}{\text{number of incudent light}} \times 100 \quad (4)$$

$$\eta(\%) = \frac{\left[\triangle G \text{ for } H_2O_2 \text{ formation } \left(J\,mol^{-1}\right)\right] \times [\text{formed amount of } H_2O_2 \text{ (mol)}]}{[\text{incident solar energy } (W)] \times [\text{time (s)}]} \times 100 \quad (5)$$

## Photocatalytic water oxidization

Photocatalyst (30 mg) was ultrasonically dispersed into a 0.01 M $NaIO_3$ aqueous solution (50 mL) with a pH buffer agent of $La_2O_3$ in a 100 mL sealed borosilicate glass vessel. The above suspension was well bubbled with nitrogen gas for 45 min to remove residual $O_2$ in the dark. The photocatalytic water oxidation reaction was triggered by irradiating simulated sunlight. The evolved $O_2$ was detected using gas chromatography with a TCD detector[11].

## Isotopic experiments using $^{18}O_2$ and $H_2^{16}O$

50 mg of Co−CN@G was dispersed into 50 mL of $H_2^{16}O$ in a 100 mL borosilicate glass vessel. The sealed glass flake was bubbled with Helium atmosphere for 30 min under stirring, $^{18}O_2$ gas (~99% purity) was then bubbled into the dispersion using a plastic syringe. The $^{18}O_2$-saturated solution was irradiated with different light sources for 15 h. The resulting reaction solution was subsequently bubbled with a Helium atmosphere for 45 min to remove the residual $^{18}O_2$ gas. Finally, 30 mL of the solution was transferred into another sealed flask containing $MnO_2$ and Helium gas using a plastic syringe, and the generated gas via the decomposition of the solution was monitored by a GC−MS system. The detected $^{16}O_2$ originated from the air when the decomposed $^{18}O_2$ gas was injected. For the triphenylphosphine (PPh$_3$) capping experiment, the filtered solution after the reaction was bubbled with $N_2$ for 2 h. In a glove box, 2 mL of solution was added to 20 mL of PPh$_3$ solution (5 mM in acetonitrile) and co−incubated in the dark for 3 h. The resulting solution was then analyzed by GC−MS system.

## Isotopic experiments with $H_2^{18}O$

4 mg Co−CN@G was dispersed into 10 mL $O_2$-saturated $H_2^{18}O$ in a 20 mL sealed glass vial. The photocatalytic reaction was performed for 2 h under different photoirradiation. The gas produced in the headspace of the sealed glass vial was detected by GC−MS system. In addition, the resulting reaction solution was also bubbled with a Helium atmosphere for 60 min in the dark to remove the $O_2$ gas in the solution. Then, 20 mL solution was transferred into another sealed glass vial containing $MnO_2$ and Helium gas by a plastic syringe, and the evolved gas was analyzed by GC−MS system[57].

## Photoelectrochemical or electrochemical measurements

Electrochemical impedance spectroscopy and the photocurrent response analysis were conducted in a three-electrode electrochemical system (CHI-660E, Pt wire counter electrode, Ag/AgCl reference electrode) under simulated solar illumination. The photocatalysts deposited on the FTO glass ($1 \times 2$ cm in size) was utilized as working electrode, which was fabricated according to the following steps: the

photocatalyst (10 mg) was fully dispersed into a mixture containing isopropanol (1 mL), and Nafion (10 μL, 5 wt%) through ultrasonic treatment to obtain a homogenous ink. Subsequently, the ink was deposited onto the cleaned FTO substrates by dip-coating with an active area of 1 cm$^2$. The photoelectrodes were then dried at 50 °C in a vacuum oven. $Na_2SO_4$ or KCl aqueous solution (0.1 M) served as the electrolyte. Before the photo-illumination, $N_2$ was continuously bubbled into the electrolyte for 40 min. The deposited photocatalyst was irradiated from the back side of the FTO electrode. Rotating disk electrode analysis was conducted on a Pine Research Instrumentation in $O_2$-saturated 0.1 M phosphate buffer solution (Pt counter electrode, Ag/AgCl reference electrode). The scan rate was set to be 5 mV s$^{-1}$. The electron transfer number for $O_2$ reduction was reckoned according to Koutecky−Levich Eq. (6)[58]:

$$\frac{1}{J} = \frac{1}{J_k} + \frac{1}{0.62nFC_0D_0^{2/3}v^{-1/6}\omega^{1/2}} \quad (6)$$

$J$ is tested current density, $J_k$ is kinetic current density, $v$ is kinetic viscosity of water, F is the Faraday constant, $\omega$ is rotating speed, $C_0$ is $O_2$ concentration in water, and $D_0$ is the $O_2$ diffusion coefficient of $O_2$.

## Calculation details

Density functional theory calculations were realized based on the CP2K code[59]. To model the extended systems, Gaussian and Plane Wave strategies were adopted[60]. The generalized gradient approximation based on Perdew-Bruke−Ernzerhof functional and the Goedecker−Teter−Hutter pseudopotentials were utilized to deal with the exchange-correlation energy[61,62]. The periodic boundary conditions in all directions have been exerted for all model systems. The adsorption energy $E_{ads}$ was determined according to $E_{ads} = E_{AB} - E_A - E_B$ ($E_{AB}$ is the energy of adsorbed structure, $E_A$ is the energy of adsorbate, and $E_B$ is the energy of adsorbent).

## Characterization

The microstructure of the photocatalysts was investigated by a Hitachi S-4800 field emission SEM and a JEOL-2010 TEM. XRD was examined on a Shimadzu XRD-6000 diffractometer (Cu Kα, 1.54178 Å). AC-HAADF-STEM was performed on a JEM-ARM200F instrument. The atomic and electronic structures of samples were investigated by XAFS measurements in the BL14W1 station (Shanghai Synchrotron Radiation Facility). Magic-angle-spinning NMR analysis was conducted on a Bruker Avance-III spectrometer. The elemental Co amount was detected on an Optima 7300DV ICP-OES. XPS was conducted on an ESCALAB 250 spectrometer. The specific surface area was characterized by adsorption-desorption analysis at 77 K and determined according to the Brunauer-Emmett-Teller method. The absorption of photocatalysts was implemented on a Cary 300 spectrophotometer. The PL spectra were collected on a JASCO FP-6500 spectrofluorometer (excitation wavelength: 375 nm).

## Data availability

The data that support the finding of this study are available from the corresponding author upon request. Source data are provided in this paper.

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

## Acknowledgements

This research is financially supported by the Hainan Science and Technology Major Project (ZDKJ2020011 awarded to N.W.), National Natural Science Foundation of China (52172195 awarded to Q.L.), the Start-up Research Foundation of Hainan University [KYQD(ZR)1907 awarded to Q. L.] and the Innovation Platform for Academicians of Hainan Province (HD-YSZX-202007 awarded to N.W., HD-YSZX-202008 awarded to Y.Y.). Q. Song thanks China Scholarship Council for the financial support of her Ph.D. study.

## Author contributions

Q.L. and N.W. led this research. W.W. and Q.L. synthesized, characterized, and analyzed the data. W.W. and Q.S. designed some contrast experiments. J.L., Y.L., and L.L. carried out the SEM and TEM characterizations, and X.H. and S.C. contributed to the photocatalytic tests. W.W. and Q.L. wrote the paper with support from K.Z. and X.D. S.S. and Y.Y. commented on the manuscript. All authors contributed to the discussion and gave approval to the final version of the paper.

## Competing interests

The authors declare no competing interests.
