## [Peer review file · Nature Communications]

REVIEWER COMMENTS

Reviewer #1 (Remarks to the Author):

The authors present cobalt single-atom catalysts loaded onto sulfur-doped graphitic carbon nitride for H₂O₂ synthesis. This material was further composite with reduced graphene oxide that exerts photothermal effects. C₃N₄ material has been widely used as a benchmark for photocatalytic H₂O₂ synthesis. Co single-atom-loaded C₃N₄ is also not new for photocatalytic H₂O₂ synthesis. Using carbonaceous material for photothermal effect and to enhance redox catalysis is a well-established subject. C₃N₄ composite with rGO is also not new to the literature. Finally, photocatalytic H₂O₂ generation in natural seawater has been already reported in the literature with pretty similar observations and discussions on the impact of salts in enhancing H₂O₂ synthesis. Given all these, I have a hard time recognizing the novelty of this work. The novelty of this work is likely related to the fact that the authors used a very particular combination of the materials, i.e., Co-C₃N₄/rGO (referred to as Co-CN@G in this study) for H₂O₂ synthesis in seawater with slightly more improved kinetics due to well-known photothermal effect. I consequently conclude that the scientific novelty and the impact on fundamental science are marginal.

As for the core mechanism, I am not convinced that Co functions as an ORR center. If so, what is water/hydroxide oxidation site? What's the proportion of H₂O₂ generated from oxygen reduction versus water oxidation? What's the quantum efficiency for each?

Despite a couple of papers on the synthesis of H₂O₂ from seawater, I am skeptical about its practicality. How do you separate H₂O₂ downstream, especially from salt? Seawater contains not only salts but also organics and microorganisms that will eventually foul the catalysts, especially when the material is carbonaceous such as rGO used in this study. One could study H₂O₂ synthesis in seawater for the sake of scientific interest, but that's already been published elsewhere.

What is the reason for improved H₂O₂ synthesis in the presence of cations? The authors mentioned electron sink (based on the similar discussion found in references cited here). But I would argue that such will (1) reduce the ORR quantum yield and (2) lead to metal reduction and catalyst surface passivation over time.

Reviewer #2 (Remarks to the Author):

Hydrogen peroxide is a powerful industrial oxidant and emerging carbon-neutral liquid fuel. Wang and coworkers constructed a cooperative photothermal-photocatalytic Co single atom anchored heterostructure for hydrogen peroxide photosynthesis from seawater. The single atom material obtained a high conversion efficiency of ~0.7% for hydrogen peroxide synthesis, outperforming the reported particulate photocatalysts in freshwater and seawater. The authors have provided appropriate characterization analysis of the materials and proper experimental control group setup for the photothermal contribution. The theoretical calculations were also performed to demonstrate the feasibility and rationality of the material's design. In my opinion, this work is very interesting. Therefore, I recommend the manuscript can be accepted to be published by Nature Communications after minor revisions.

1. Hydrogen peroxide is not very stable under UV light soaking. The authors evaluated the decomposition rate by fitting the time course of hydrogen peroxide synthesis. The decomposition behavior should be examined by dispersing photocatalysts into hydrogen peroxide solution under simulated solar irradiation (no filter). Also, how about the decomposition at different temperatures.
2. In Fig. 2b, the transient photocurrent response of Co-CN@G photocatalysts increased with the extension of light soaking, and remained almost constant after about 2000 s. The authors should give a reasonable explanation for such enhancement.
3. The photocatalysts have a potential concern of their long-term stability since the aggressive hydroxyl radical and superoxide radical are concomitantly generated during hydrogen peroxide photosynthesis. The catalysts may show some oxidation on their surface or were damage after continuous photocatalysis, FTIR spectroscopy measurement after photocatalysis should be performed. Moreover, the Co leaching experiments across the reaction time should be included. Also, to highlight the strength of this work, the authors should also evaluate the stability of pristine Co-CN or reference Co-CN@G, and provide a reasonable explanation to clarify the difference in the manuscript.
4. The authors stated that the excessive amounts of either Co, Co-CN or RGO resulted in an inferior photocatalytic activity, please provide detailed explanation. Co nanoparticle/nanocluster agglomeration could be observed at higher Co loading? Whether the specific surface area and pore distribution significantly affected the catalytic performance? The data need be provided.
5. The photocatalysts contained graphene oxide component in their structure. It is well-known that the carbon materials could be aggregated in high salt seawater because of the electric double layer suppression. Such agglomeration has been observed in this work? Please clarify it.
6. The authors stated that hydrogen peroxide can be produced through oxygen reduction reaction and water oxidation reaction. The latter route was identified by isotopic labeling experiments. In general, water oxidation needs a high potential, theoretically at 1.76 V versus NHE, which has rarely been mentioned in previous studies. It would be important for the reader that the authors should include the energy level of the photocatalysts in the schematic figure, Fig. 5c. Also, the authors should give a brief description on the formation of hydrogen peroxide in the context.

7. The detailed information on the outdoor test should be provided in method section, such as the oxygen partial pressure, solution volume, etc. The figure labels in the inset of Fig. 3i and Fig. 4c are difficult to read. The grammar errors and mistakes in this manuscript need be corrected carefully.

Reviewer #3 (Remarks to the Author):

Wang et al. report on solar H₂O₂ production using C₃N₄ functionalized with Co single atom catalysts. This is an important research question, especially since most photocatalyst get deactivated in seawater. Despite the impressive performance of the catalyst, there several issues that need to be addressed, before the manuscript could be considered for publication.

1) The authors present a model for theoretical studies on the catalytic activity with assumed single atom precision (Co). However, the theoretical model lacks the experimental basis, because the authors have determined the nitrogen positions with XPS, which does not tell whether one has N-clustering or not (similar to bipyridine). The N positions determine the bonding of Co and also have an influence on the electronic structure. This can be solely determined according with ¹⁵N solid state NMR spectroscopy. Despite the low abundancy of ¹⁵N, recent reports have shown that it is relatively easy to obtain solid ¹⁵N ssNMR spectra (from un-labelled N-carbon) if the N content is above ca. 3.5%

(Chem. Mater. 2020, 32, 17, 7263–7273, <https://doi.org/10.1021/acs.chemmater.0c01666> and

ACS Catal. 2021, 11, 22, 14087–14101, <https://doi.org/10.1021/acscatal.1c03733>). The first paper reports the method and the second even applies it to C₃N₄ photocatalysts for H₂O₂ production, which is as such the ideal reference for the material of the current manuscript. The N position could be also determined for a sample without Co in order to get the correct model for the computational studies. XPS is not the state-of-the-art to determine the type of nitrogen in the material. Atomic precision is required if one wants to set up a model. In addition, the nitrogen species also affect the adsorption of the substrate.

2) Fig 1 b shows the TEM EDX mapping, but there is no information on the chemical composition. What is the C/N ratio?

3) Supplementary table S1 is not mentioned in the manuscript and it states the current catalyst to have the highest activity. One cannot really compare a catalyst that is based on heavier elements with a catalyst that is mainly composed of carbon and nitrogen. This should be rather related to activity/mole. I understand that this is quite difficult, but a sentence should be mentioned here.

4) The ¹⁸O₂ experiments still show lots of ¹⁶O₂ and the authors claim that this stems from oxygen which is present in air during sample preparation. I agree that the ¹⁸O₂ tests are a proof, but I would

suggest to make one additional test as with a phosphine capping agent and then to detect the m/z value.

5) The authors write:

“In summary, we reported a photothermal-photocatalytic Co-CN@G particulate photocatalyst for highly efficient non-sacrificial H₂O₂ production from natural seawater. By taking advantage of the cooperation of CN@G heterostructure and isolated Co single atoms, this system exhibited excellent and stable catalytic activity towards H₂O₂ photosynthesis”. If you take a look at Fig3b, then you see that the catalyst gets deactivated after 6h significantly. I am not that convinced that one can speak here about a stable catalyst in sea water; this resembles rather the previously published catalyst. I would recommend to re-phrase the claim of the manuscript. Otherwise, it is a bit misleading regarding the stability.

Dear Editors and Reviewers:

Thank you for your letter and for the reviewers' comments concerning our manuscript entitled "Photothermal-Enabled Single-Atom Catalysts for High-Efficiency Hydrogen Peroxide Photosynthesis from Natural Seawater" (Full Paper, NCOMMS-22-40934). The invited major revision of this manuscript has been finished with significant additional experimental data. We are sorry that it took us a relatively long time to prepare the revised manuscript since COVID spreads across China (campus closed) and the Spring Festival holiday. These comments are all valuable and very helpful for revising and improving our paper. We have studied comments carefully and have made correction which we hope meet with approval. Revised portions are marked in red in the revised manuscript. The main corrections in the paper and the response to the reviewer's comments are as following:

Reviewer #1 (Remarks to the Author):

The authors present cobalt single-atom catalysts loaded onto sulfur-doped graphitic carbon nitride for H₂O₂ synthesis. This material was further composite with reduced graphene oxide that exerts photothermal effects. C₃N₄ material has been widely used as a benchmark for photocatalytic H₂O₂ synthesis. Co single-atom-loaded C₃N₄ is also not new for photocatalytic H₂O₂ synthesis. Using carbonaceous material for photothermal effect and to enhance redox catalysis is a well-established subject. C₃N₄ composite with rGO is also not new to the literature. Finally, photocatalytic H₂O₂ generation in natural seawater has been already reported in the literature with pretty similar observations and discussions on the impact of salts in enhancing H₂O₂ synthesis. Given all these, I have a hard time recognizing the novelty of this work. The novelty of this work is likely related to the fact that the authors used a very particular combination of the materials, i.e., Co-C₃N₄/rGO (referred to as Co-CN@G in this study) for H₂O₂ synthesis in seawater with slightly more improved kinetics due to well-known photothermal effect. I consequently conclude that the scientific novelty and the impact on fundamental science are marginal.

Response: Thanks for your value comments.

Photocatalytic H₂O₂ synthesis have gained considerable research momentum in recent years. For a practical photocatalytic process, the pursuit of high solar conversion efficiency is the most important research topic. Unfortunately, the previously reported photocatalysts generally suffer from low solar-to-H₂O₂ conversion efficiency in pure water (~0.6%) and seawater (~0.2%). Cobalt-based materials, including Co single atom catalysts, are regarded as one of the most active metal-based catalysts for O₂ reduction in electrochemical reaction (Xie, X. et al. *Nat. Catal.* **3**, 1044-1054 (2020); Smith, R. L. et al. *Nat. Commun.* **8**, 2022 (2020)). It is true that a well-designed Co single-atom-loaded C₃N₄ photocatalyst with spatial separation of oxidative and reductive sites has been reported for photocatalytic H₂O₂ synthesis in pure water (Chu, C. et al. *Proc. Natl. Acad. Sci. U.S.A.* **117**, 6376-6382 (2020)). Nevertheless, H₂O₂ generation is very low under AM 1.5G irradiation (only 0.25 mM after 8 h). The most critical reason for the poor catalytic activity of Co-based semiconductor is the intermediate bands formed by the half-filled *d* electrons (Gao, J. et al. *Chem* **6**, 1-17 (2020); Jung, E. et al. *Nat. Mater.* **19**, 436-442 (2020)), which will lead to the serious carrier recombination and thus the low solar conversion efficiency. In this context, we dedicate our study to overcoming the charge recombination and increasing the conversion efficiency by rational material design and sufficient incident light utilization. The advances of our work could be concluded as follows: 1) Novel material: The Co-CN@G with the particular combination in this study is a novel photocatalyst for H₂O₂ photosynthesis. Co single atoms grown on sulfur doped C₃N₄/RGO 2D heterojunction with intimate electronic coupling address the serious recombination of photogenerated carriers. Such strategy is critical for obtaining the high catalytic activity. 2) Photothermal effect: Although using carbonaceous material for photothermal effect to enhance redox catalysis is well-established in some photocatalytic reactions, no report demonstrating the use of photothermal effect is presented for photocatalytic H₂O₂ synthesis. The photothermal effect yielded about 22% enhancement in H₂O₂ synthesis under simulated solar irradiation. Such improvement is obvious and significant. The direct integration of photothermal-photocatalytic heterostructure system represents a promising direction for developing efficient photo-response materials for H₂O₂ production. 3) Record-high

efficiency: The H₂O₂ amount formed on Co-CN@G photocatalyst is 7.74 mM after 6 h reaction, which is about 30 times higher than that of Co single-atom-loaded C₃N₄ in previous study (0.25 mM after 8 h). Co-CN@G yielded a record-high solar-to-H₂O₂ conversion efficiency (0.72%) under simulated sunlight irradiation (AM 1.5G), which is higher than that of the state-of-the-art photocatalysts in pure water, and 2.4 times higher than that of photocatalyst in seawater (Supplementary Table 2). Moreover, the H₂O₂ production of Co-CN@G under outdoor natural sunlight irradiation also achieved a promising daily yield of 8508 μmol g⁻¹. 4) New reaction pathway: We show that the Co-CN@G enables a new two-electron water oxidation pathway toward photocatalytic H₂O₂ production. In sharp contrast, this reaction pathway was not able to be observed on the pristine Co-CN and CN@G samples. This is the first demonstration on the H₂O₂ generation from the two-electron water oxidation on single-atom-based photocatalysts. The study unveils that the particular combination of Co-CN@G is vital to achieve efficient H₂O₂ production from two-electron water oxidation pathway. In addition, we also found a new pathway for promoting four-electron water oxidation in this revised manuscript. The halide ions (Cl⁻ and Br⁻) in seawater can be oxidized by photogenerated holes, assisting four-electron water oxidation and promoting the overall H₂O₂ synthesis. Overall, the results in this study demonstrate the possibility of realizing high-efficiency H₂O₂ photoproduction from seawater.

1. As for the core mechanism, I am not convinced that Co functions as an ORR center. If so, what is water/hydroxide oxidation site? What's the proportion of H₂O₂ generated from oxygen reduction versus water oxidation? What's the quantum efficiency for each?

Response: Thanks for your value comments.

The *d* orbitals of Co metal can interact with the *p* electrons of the oxygen feedstock and intermediates during O₂ reduction process, which facilitates the adsorption of O₂ molecules and the subsequent electron transfer (Zhao, C. et al. *Angew. Chem. Int. Ed.* **60**, 4448-4463 (2021); Wang, X. et al. *Chem* **5**, 1486-1511 (2019)). In fact, Co-based single atom catalysts are particularly attractive for electrochemical O₂ reduction

reaction, and the Co centers are typically regarded as the reduction active sites of O₂ (Zhao, C. et al. *Angew. Chem. Int. Ed.* **60**, 4448-4463 (2021); Yang, X. et al. *Adv. Mater.* **34**, 2107954 (2022); Huang, K. et al. *Nat. Commun.* **10**, 606 (2019)). In our study, Co-CN@G heterostructure yielded ~3 times of H₂O₂ amount of the pristine CN@G, indicating that Co single atom plays the most significant role in determining the catalytic activity. Based on DFT calculations, the single Co sites with the end-on type O₂ adsorption are able to accumulate electrons, which function as O₂ reduction sites. The end-on adsorption shall notably suppress the four-electron O₂ reduction, leading to a high selectivity of the two-electron process. The step-wise two-electron O₂ reduction pathway contains two proton coupled electron transfer steps with OOH* as the only intermediate. The absorption of OOH* intermediate on different sites, including Co, N, S and C sites, was studied. Significantly, OOH* prefers to bind to Co sites with a binding free energy ΔG_{OOH^*} of 3.98 eV, close to the optimum value of 4.22 eV. By contrast, the ΔG_{OOH^*} for the C site is 4.96 eV, and the absorption on N and S sites is unstable. Thus, the DFT calculations reveal that *OOH absorbed on the top of the Co sites is energetically more favorable, which is similar with the previous reports (Chen, S. et al. *J. Am. Chem. Soc.* **144**, 14505-14516 (2022); Zhao, Q. et al. *Energy Environ. Sci.* **14**, 5444-5456 (2021)). These results suggest that Co can function as the main active center for O₂ reduction. Simultaneously, the counter half reaction-water oxidation occurs at the N atoms (NC₂ site) of the melem unit adjacent to the Co site (Supplementary Fig. 36).

Supplementary Figure 36. The most stable configuration of reaction intermediates

OH* on the N site of Co-CN@G.

In our study, H₂O₂ was generated on Co-CN@G via the two-electron O₂ reduction and two-electron water oxidation (Co-CN and CN@G can only drive H₂O₂ synthesis through the single source of the two-electron O₂ reduction). In this situation, it is difficult to distinguish the proportion of H₂O₂ generated from O₂ reduction versus water oxidation because of the coupled reactions (O₂ produced from water oxidation can also be reduced into H₂O₂ by photogenerated electrons). On the one hand, H₂O₂ is known to react with triphenylphosphine (PPh₃) to give triphenylphosphine oxide (OPPh₃), we thus performed the isotopic experiments for ¹⁸O₂ and H₂¹⁸O labeling tests using PPh₃ as the capping agent to distinguish the proportion of the formed H₂O₂. The resulting OPPh₃ were then analyzed by GCMS. As demonstrated, both ¹⁶O=PPh₃ and ¹⁸O=PPh₃ were observed from the GCMS spectrum when labeled ¹⁸O₂ (Supplementary Fig. 32) was used. These results indicates that the dissolved O₂ by bubbling, the formed O₂ from four-electron water oxidation and the two-electron water oxidation contributed to the production of H₂O₂. Consequently, although the isotopic experiments were performed, the three-channel and coupled formation pathways of H₂O₂ still make it difficult to distinguish the proportion of H₂O₂ generated from O₂ reduction versus water oxidation in real situation. On the one hand, H₂O₂ produced from O₂ reduction and water oxidation were also separately investigated in the presence of electron or hole scavenger. 1.56 mM of H₂O₂ was generated after light-irradiating the N₂-saturated seawater containing KBrO₃ additive (electron sacrificial agent) (H₂O₂ was formed from water oxidation), and 10.6 mM of H₂O₂ was produced in the O₂-saturated seawater in the presence of hole scavenger methanol (H₂O₂ was generated from the reduction of dissolved O₂). In this situation, the proportion of H₂O₂ generated from the reduction of dissolved O₂ versus water oxidation was calculated to be 6.8:1. Correspondingly, the quantum efficiency at 420 nm was determined to be 1.86% and 13.02% for water oxidation and the reduction of dissolved O₂ (Fig. R1), respectively. However, this proportion did not reflect the real situation of the photocatalytic H₂O₂ synthesis since charge scavengers can significantly affect the charge separation of photocatalysts and

O₂ reduction/water oxidation. Furthermore, the O₂ generated from the four-electron water oxidation did not undergo the further reduction in the presence of hole scavenger. Therefore, the proportion of H₂O₂ generated from O₂ reduction versus water oxidation and the corresponding quantum efficiencies are not included in the manuscript.

Fig. R1 a) H₂O₂ production and b) apparent quantum yield of Co-CN@G from the half reaction of water oxidation and O₂ reduction in the presence of charge scavenger.

2. Despite a couple of papers on the synthesis of H₂O₂ from seawater, I am skeptical about its practicality. How do you separate H₂O₂ downstream, especially from salt? Seawater contains not only salts but also organics and microorganisms that will eventually foul the catalysts, especially when the material is carbonaceous such as rGO used in this study. One could study H₂O₂ synthesis in seawater for the sake of

scientific interest, but that's already been published elsewhere.

Response: Thanks for your value comments.

H₂O₂ is one of the most important chemicals in industry and widely used as a green oxidant in disinfectants, bleaching agents, sanitizing agents, and chemical synthesis. In our study, the Co-CN@G photocatalyst can produce H₂O₂ from both freshwater and seawater under light irradiation with a high solar conversion efficiency. On the one hand, the salt ions in H₂O₂-seawater solution could, in principle, be reduced or removed through conventional desalination and reverse osmosis methods (Sutherland, B. R. *Joule* **3**, 1404-1414 (2019)). On the other hand, H₂O₂ is an emerging liquid energy carrier. The energy density of aqueous H₂O₂ (60%) is 3.0 MJ L⁻¹ (2.1 MJ kg⁻¹), which is comparable to the value of compressed hydrogen (35 MPa, 2.8 MJ L⁻¹, 3.5 MJ kg⁻¹). In fact, H₂O₂ can be directly used as feedstock for fuel cells and new-generation rockets (Shaegh, S. A. M. et al. *Energy Environ. Sci.* **5**, 8225-8228 (2012); Yamada, Y. et al. *Energy Environ. Sci.* **4**, 2822-2825 (2011); Han, L. et al. *Adv. Energy Mater.* **5**, 1400424 (2015); Yang, Y. et al. *Chem. Eng. J.* **369**, 813-817 (2019); Li, M. et al. *Acta Astronaut.* **190**, 283-298 (2022); Das, J. et al. *Chem. Eng. J.* **426**, 131806 (2021); Okninski, A. et al. *Aerosp. Sci. Technol.* **82-83**, 284-293 (2018); Petrutik, N. et al. *Chem. Eng. J.* **454**, 140170 (2023)).

In energy-related fields, seawater is the most convenient medium and is of significant interest because of their abundance and sustainability. In previous reports, H₂O₂ produced from seawater can be directly used as a solar fuel in H₂O₂ fuel cell to generate electricity, with an open-circuit voltage of 0.78 V and a maximum power density of ~2 mW cm⁻² (Mase, K. et al. *Nat. Commun.* **7**, 11470 (2016); Mase, K. et al. *ACS Energy Lett.* **1**, 913-919 (2016)). In addition, the researchers have started to explore the potential of electrochemical H₂O₂ production from the simulated seawater (Zhao, Q. et al. *Energy Environ. Sci.* **14**, 5444-5456 (2021)). Although many things remain to be done on the solar-to-H₂O₂ conversion efficiency and seawater-H₂O₂ fuel cells, it is highly desired to utilize the most earth-abundant seawater instead of precious freshwater for the large-scale and economical use of H₂O₂ as a solar fuel to construct an ideal energy-sustainable society. Moreover, H₂O₂ generation from seawater can also

be directly applied in a wide variety of applications, including ballast water treatment (Wang, W. et al. *ACS EST Water* **1**, 1483-1494 (2021); Zhang, C. et al. *J. Environ. Chem. Eng.* **9**, 105973 (2021)), seawater pretreatment (Villar-Navarro, E. et al. *Sol. Energy* **177**, 144-154 (2019)), degradation of toxins and pollutants in seawater (Erick, R. B. et al. *Desalination* **245**, 135-145 (2009); Calza, P. et al. *Sci. Total Environ.* **431**, 84-91 (2012)), seawater disinfection (Rubio, D. et al. *Water Res.* **47**, 6367-6379 (2013); Penru, Y. et al. *J. Photoch. Photobio. A* **233**, 40-45 (2012)) and seawater flotation mineral system (Yang, X. et al. *Miner. Eng.* **176**, 107356 (2022)). As a solar fuel and disinfectant, the H₂O₂ production from natural seawater can not only make full use of abundant seawater resource but also greatly reduce the cost of the solar energy conversion process, which is more in line with many actual application requirements.

The raw seawater is rich in inorganic salts (Na⁺, Mg²⁺, Cl⁻, SO₄²⁻, etc.), organic matter, bacteria/microorganisms, as well as small particulates. On the one hand, the natural seawater for H₂O₂ photosynthesis in our study has been filtered before use through a filter paper (pore size: 4-5 μm) and a polymer PTFE membrane filter (pore size: 0.22 μm) to remove suspended matter and microorganisms, respectively. It is true that the existence of interfering ions, bacteria and microorganisms in seawater may poison or corrode the electrocatalysts and photocatalysts for hydrogen production from seawater splitting and limit their long-term stability. However, photocatalytic H₂O₂ synthesis from seawater is obviously distinguished from the photocatalytic, photoelectrochemical and electrocatalytic hydrogen production from seawater splitting in terms of anti-interfering ability. The salt ions in seawater can promote the H₂O₂ synthesis for some special photocatalysts, as demonstrated by Co-CN@G in this work and the previous reports (Wu, Q. et al. *Nat. Commun.* **12**, 483 (2021); Gopakumar, A. et al. *J. Am. Chem. Soc.* **144**, 2603-2613 (2022)). These results indicate that the effect of salt ions is strongly dependent on the material's composition. On the other hand, in photocatalytic H₂O₂ synthesis, the formed reactive oxygen species (ROS), including H₂O₂, hydroxyl radical intermediate (•OH) and superoxide intermediate (•O₂⁻), are strong oxidants and can degrade the organic matter and inactivate the bacteria in seawater. As reported previously, photocatalysis in advanced oxidation processes has

been regarded as a reliable strategy for combating biofilm and marine biofouling, the generated ROS can disinfect pathogens and inactivate bacteria/microorganisms by damaging essential macromolecules (Liu, C. et al. *Nat. Nanotechnol.* **11**, 1098-1104 (2016); Luo, Q. et al. *Adv. Sci.* **9**, 2105346 (2022)). In contrast, the seawater splitting for hydrogen production via electro/photoelectrochemical catalytic reactions does not involve the production of ROS, thus the organic matter and microorganism in seawater could deactivate the catalyst materials. Moreover, it was found in this revised manuscript that the Cl^- can be oxidized by photogenerated holes. The resulting HClO/ClO^- could also serve as the antibiofouling agent to prevent the biofouling. For instance, in a 13.3 MGD (million gallon per day) seawater desalination plant located in Yanbu industrial city, seawater is treated by injecting sodium hypochlorite (NaOCl) generated on-site by the electrolysis of seawater to remove algae and bacteria and to prevent microorganism growth (Khawaji, A. D. et al. *Desalination* **203**, 176-188 (2007)). In addition, as an efficient chemical clean technique in industrial reverse osmosis membrane for desalination, H_2O_2 or NaOCl was also typically used to remove the fouling organic matter and microorganisms. In our study, the stability evaluation confirms that Co-CN@G can keep active after multiple cycles. Thus, although the composition of natural seawater is complicated, the practical viability of H_2O_2 production from seawater is not seriously challenged by salt- and bio-fouling encountered in other processes using seawater as the feed.

3. What is the reason for improved H_2O_2 synthesis in the presence of cations? The authors mentioned electron sink (based on the similar discussion found in references cited here). But I would argue that such will (1) reduce the ORR quantum yield and (2) lead to metal reduction and catalyst surface passivation over time.

Response: Thanks for your value comments.

We understand your concern that the large number of ions in seawater could reduce the photocatalytic performance due to the metal reduction and catalyst surface passivation over time. Many reports on electrocatalytic/photoelectrocatalytic seawater splitting for hydrogen production claimed that the dissolved salts may deactivate

catalysts, consume the photogenerated carriers and lead to undesirable side reactions (Guo, J. et al. *Nat. Energy* (2023) <https://doi.org/10.1038/s41560-023-01195-x>; Dong, W. et al. *Nat. Commun.* **14**, 179 (2023)). From another perspective, the inorganic ions in seawater may also play the role of promoters or sacrificial agent to improve the photocatalytic performance of hydrogen production from seawater splitting (such as La₂NiO₄: Ma, X. et al. *Mater. Today Nano* **21**, 100289 (2023); brookite TiO₂/Pt: Zhang, J. et al. *Nano Res.* **15**, 2013-2022 (2022)).

On average, the natural seawater has a salinity of 3.1~3.5%. It contains various ions such as Na⁺, K⁺, Mg²⁺, Ca²⁺, Cl⁻, Br⁻, SO₄²⁻, and CO₃²⁻, with NaCl being the predominant component (Millero, F. J. et al. *Deep-Sea Res. PT. I* **55**, 50-72 (2008); Dresp, S. et al. *ACS Energy Lett.* **4**, 933-942 (2019); *Comprehensive Handbook of Iodine, Academic Press*, 83-91 (2009)). These ions compose more than 99.9% of salinity in seawater. The mean chemical composition of seawater species and the corresponding standard redox potential were given as following (Dresp, S. et al. *ACS Energy Lett.* **4**, 933-942 (2019)). According to the listed potentials and chemical composition, the main metal cations in seawater are very hard to be reduced by the photogenerated electrons due to the high reduction potential.

Cations:

Mn: 3.6×10^{-9} mol/kgH₂O, Ag: 4×10^{-7} mol/kgH₂O, Au: 2×10^{-7} mol/kgH₂O, Co: 9×10^{-7} mol/kgH₂O, Cr: 5.8×10^{-9} mol/kgH₂O, Cs: 3×10^{-9} mol/kgH₂O, Cu: 7.8×10^{-8} mol/kgH₂O, Fe: 4×10^{-8} mol/kgH₂O, Ni: 3×10^{-8} mol/kgH₂O, Pb: 1.9×10^{-10} mol/kgH₂O.

Anions:

SO_4^{2-} : 0.02927 mol/kg H_2O , $2\text{SO}_4^{2-} \rightarrow \text{S}_2\text{O}_8^{2-} + 2\text{e}^-$, $E^0 = 2.010 \text{ V}_{\text{SHE}}$

F^- : 0.00007 mol/kg H_2O , $\text{F}^- \rightarrow \text{F}_2 + 2\text{e}^-$, $E^0 = 2.866 \text{ V}_{\text{SHE}}$

HCO_3^{2-} : 0.00183 mol/kg H_2O , CO_3^{2-} : 0.00027 mol/kg H_2O , OH^- : 0.00001 mol/kg H_2O ,
 I^- : 3.9×10^{-7} mol/kg H_2O .

In addition, we have performed the XPS measurements to investigate the oxidation states of Co of Co-CN@G (Fig. R2) and the main absorbed cations (Na^+ , K^+ , Mg^{2+} , Ca^{2+}) on the surface of Co-CN@G (Fig. R3) before and after 15 photocatalytic cycles. No obvious change of the oxidation states of these metal cations was observed. Furthermore, the stability evaluation reveals that Co-CN@G in this work can keep active after multiple cycles. The unchanged XRD pattern, FT-IR spectra, and inapparent morphology transformation also confirmed the robust nature of Co-CN@G (Supplementary Fig. 25).

Fig. R2 High-resolution XPS spectra of Co 2p before and after photocatalytic cycles.

Fig. R3 High-resolution XPS spectra of the main metal cations in seawater before and after photocatalytic cycles. a) Na 1s, b) K 2p, c) Mg 1s, and d) Ca 2p.

In our study, it was clearly observed that Co-CN@G showed a significantly higher catalytic activity for H_2O_2 photosynthesis in natural seawater than in pure water. Such enhancement for H_2O_2 generation in natural seawater should be caused by the various dissolved cations and anions of seawater. In this revised manuscript, we have performed a series of additional experiments to clarify the possible mechanisms for the improved H_2O_2 synthesis in the presence of seawater ions. First, it was observed from Supplementary Fig. 26 in our preliminary study that the individual addition of the NaCl, MgCl_2 , CaCl_2 , KCl, Na_2SO_4 and Na_2CO_3 into deionized water increased H_2O_2 evolution over Co-CN@G. It has been reported that the electron sink of oxygen-containing groups of carbon materials in the presence of metal cations could boost photocatalytic activity by attracting electrons (Wu, Q. et al. *Nat. Commun.* **12**, 483

(2021)). To understand the synthetic effect of seawater ions on the overall photocatalytic H_2O_2 production more comprehensively, simulated seawater was also prepared by dissolving the chemicals (NaCl , 27.21 g; MgCl_2 , 3.81 g; MgSO_4 , 1.66 g; CaSO_4 , 1.40 g; K_2SO_4 , 0.58 g; K_2CO_3 , 0.21 g; MgBr_2 , 0.08 g) in 1 L of deionized water. As added in Fig. 3b in this revised manuscript, the natural seawater and simulated seawater show a similar H_2O_2 generation under the same conditions, implying that the composition of the simulated seawater is very close to those of the natural seawater. Then, photocatalytic H_2O_2 synthesis from simulated seawater with different salinities was investigated (Supplementary Fig. 27). Salinity of prepared simulated seawater was varied from 0.2 to 3.4%. We observed a volcanic relationship between H_2O_2 evolution and salinity with a summit at a moderate salinity of 2.4%. It implies that H_2O_2 production will be restrained by high salt concentration and can be enhanced by pretreatment of mild desalination.

Supplementary Figure 27. Photocatalytic H_2O_2 synthesis from simulated seawater with different salinities.

To distinguish the effect of cations on H_2O_2 production, one cation was removed from the simulated seawater. To keep the same ion concentration, ammonium chloride or sulfate was added (Supplementary Table 3). Fig. 3i presents the H_2O_2 production in simulated seawater in the absence of a single Na^+ (NH_4Cl was used to replace NaCl),

K^+ ($(NH_4)_2SO_4$ was used to replace K_2SO_4 and K_2CO_3), Mg^{2+} (NH_4Cl was used to replace $MgCl_2$, $(NH_4)_2SO_4$ was used to replace $MgSO_4$, NH_4Br was used to replace $MgBr_2$), or Ca^{2+} cation ($(NH_4)_2SO_4$ was used to replace $CaSO_4$). We found that the photocatalytic activity was suppressed in the absence of Na^+ and Mg^{2+} , while the removal of K^+ and Ca^{2+} slightly reduced H_2O_2 formation because their actual concentrations in seawater are low (K^+ : 0.009 M, Ca^{2+} : 0.01 M). Based on these results, it could be concluded that the existence of metal cations in natural and simulated seawater favors H_2O_2 photosynthesis, which is similar with the individual addition of inorganic metal salt.

Fig. 3i Photocatalytic H_2O_2 synthesis from simulated seawater in the absence of different metal cations.

To understand the promotional role played by different metal cations in the O_2 reduction process, DFT calculations were performed to understand the effect of cations on charge transfer (Supplementary Fig. 28). Hydrated cations were considered to mimic the experimental environment during the calculations, with the optimized number of water molecules bound to Na^+ and Mg^{2+} being three and four, respectively. Interestingly, the interlayer charge transfer of Co-CN@G was improved after interacting with $Na^+(H_2O)_3$ or $Mg^{2+}(H_2O)_4$ cation. These results indicate that Na^+ and Mg^{2+} is beneficial

to the charge separation/migration of photogenerated carriers. This trend was also confirmed experimentally by resorting time-resolved PL spectroscopy (Supplementary Fig. 29). Co-CN@G treated with different metal chlorides exhibited the increased intensity-average PL lifetime (τ), indicating that the presence of metal cations can improve the efficiency of photoexcited carriers, which is consistent with the DFT calculations.

Supplementary Figure 28. The Mulliken charge difference between each adjacent layers of Co-CN@G for the enlarged cross view in the presence of a) $\text{Na}^+(\text{H}_2\text{O})_3$ and b) $\text{Mg}^{2+}(\text{H}_2\text{O})_4$.

Supplementary Figure 29. Time-resolved PL spectra of Co-CN@G in different metal chlorides solutions (0.05 M).

Taking into account all anions with their corresponding standard redox potentials given above, the oxidation of bromide ($\text{Br}^- \rightarrow \text{Br}_2 + 2\text{e}^-$, $E^0 = 1.087 \text{ V}_{\text{SHE}}$) and chloride ($\text{Cl}^- \rightarrow \text{Cl}_2 + 2\text{e}^-$, $E^0 = 1.36 \text{ V}_{\text{SHE}}$) could compete with the oxidation of water ($\text{H}_2\text{O} \rightarrow 2\text{H}^+ + 0.5\text{O}_2 + 2\text{e}^-$, $E^0 = 1.23 \text{ V}_{\text{SHE}}$). To determine whether the anions affect the photocatalytic process, we first examined the photocatalytic activity in the presence of various sodium salts. The comparison experiments were performed in 0.05 M NaCl, NaBr, NaF, and NaNO_3 for 10 h, respectively. As demonstrated in Supplementary Fig. 30, it is observed that H_2O_2 generation in NaCl or NaBr solution is higher than that in NaF and NaNO_3 solutions. This result indicates that the halide anions (Cl^- and Br^-) can exert a positive effect on the photocatalytic reactions. Similarly, when both Cl^- and Br^- were absent from simulated seawater (NaF was used to replace NaCl, $\text{Mg}(\text{NO}_3)_2$ was used to replace MgCl_2 and MgBr_2), the H_2O_2 production decreases from 8.78 mM to 8.32 mM due to the avoidance of halide ions. The effect of low-concentration Br^- (0.00087 M) can be neglected in simulated seawater. Thus, it is further confirmed that halide ions (mainly Cl^-) play a critical role in the enhancement of H_2O_2 production. In fact, although the oxidation potential of Cl^- to Cl_2 is more thermodynamically unfavorable than the four-electron water oxidation, it is kinetically favorable for Cl^- oxidation by photogenerated holes in the valence band of the Co-CN@G due to an only two-electron redox process. The water oxidation enables a local acidification on the surface of Co-CN@G photocatalyst. The generated Cl_2 could react with water to form hypochlorous acid (HClO) ($\text{Cl}_2 + \text{H}_2\text{O} \rightarrow \text{H}^+ + \text{Cl}^- + \text{HClO}$), which would be easily decomposed into Cl^- and O_2 under light irradiation ($\text{HClO} \rightarrow \text{H}^+ + \text{Cl}^- + 1/2\text{O}_2$). During this process, the overall reaction is expressed as following: $\text{Cl}^- + \text{H}_2\text{O} \rightarrow 2\text{H}^+ + \text{Cl}^- + 0.5\text{O}_2 + 2\text{e}^-$. Thus, the oxidation of halide ions actually boosts the four-electron water oxidation. Notably, the O_2 amount generated in N_2 -saturated NaIO_3 aqueous solution containing NaCl additive after 2 h (3.82 μmol) was 0.3 times higher than that with NaF additive (Supplementary Fig. 31), further verifying that Cl^- oxidation boosts the water oxidation. The low concentration of Br^- (0.00087 mol/kg H_2O) made its oxidation typically neglected as a first approximation in comparison with Cl^- oxidation.

Overall, although no empirical study can provide a definitive answer to the

improved H_2O_2 photosynthesis from seawater caused by cations and anions, we believe our findings make an important contribution to advancing our understanding of the topic.

Supplementary Figure 30. a) H_2O_2 photoproduction in different solutions (0.05 M) for 10 h. b) Photocatalytic H_2O_2 synthesis from simulated seawater for 10 h in the absence of Cl^- and Br^- .

Supplementary Figure 31. Time course of photocatalytic O_2 evolution over Co-CN@G in NaCl and NaF solution (0.05 M) during the half reaction.

Reviewer #2 (Remarks to the Author):

Hydrogen peroxide is a powerful industrial oxidant and emerging carbon-neutral liquid

fuel. Wang and coworkers constructed a cooperative photothermal-photocatalytic Co single atom anchored heterostructure for hydrogen peroxide photosynthesis from seawater. The single atom material obtained a high conversion efficiency of ~0.7% for hydrogen peroxide synthesis, outperforming the reported particulate photocatalysts in freshwater and seawater. The authors have provided appropriate characterization analysis of the materials and proper experimental control group setup for the photothermal contribution. The theoretical calculations were also performed to demonstrate the feasibility and rationality of the material's design. In my opinion, this work is very interesting. Therefore, I recommend the manuscript can be accepted to be published by Nature Communications after minor revisions.

Answer: Thanks for your inspiring comments and recommendation. Following the reviewer's suggestions, we have carefully and thoroughly revised the manuscript, and the changes made in the manuscript are highlighted in red.

1. Hydrogen peroxide is not very stable under UV light soaking. The authors evaluated the decomposition rate by fitting the time course of hydrogen peroxide synthesis. The decomposition behavior should be examined by dispersing photocatalysts into hydrogen peroxide solution under simulated solar irradiation (no filter). Also, how about the decomposition at different temperatures.

Response: Thanks for your value comments.

It's true that H_2O_2 is not very stable under long-term UV light soaking due to its decomposition. Following the reviewer's suggestion, we have evaluated the decomposition behavior by dispersing photocatalysts into hydrogen peroxide solution under simulated solar irradiation (no filter) in this revised manuscript. As shown in Supplementary Fig. 22, the 60 min decomposition of H_2O_2 over Co-CN@G sample (14.1%) was lower than that on the pristine CN@G (20.0%) or Co-CN (23.5%) sample, indicating the suppressed decomposition of H_2O_2 on Co-CN@G. In addition, the H_2O_2 is relatively stable at different temperatures in the presence of Co-CN@G under dark, revealing that the temperature exerts a negligible effect on the H_2O_2 decomposition below 50 °C.

Supplementary Figure 22. a) The photocatalytic decomposition of H₂O₂ (C₀ = 1 mM) over Co-CN@G, CN@G, and Co-CN samples. b) The decomposition of H₂O₂ at different temperatures in the presence of Co-CN@G.

2. In Fig. 2b, the transient photocurrent response of Co-CN@G photocatalysts increased with the extension of light soaking, and remained almost constant after about 2000 s. The authors should give a reasonable explanation for such enhancement.

Response: Thanks for your value comments.

Co-CN@G presented a significant response of the photocurrent and the value gradually increased until reaching a steady state with the extension of light illumination. We hypothesize that the photothermal effect can increase the electrolyte temperature, which can promote the photogenerated charge separation and accelerate carrier migration. The similar trend was also observed in the previous report (Dai, B. et al. *Adv. Mater.* **32**, 1906361 (2020)).

3. The photocatalysts have a potential concern of their long-term stability since the aggressive hydroxyl radical and superoxide radical are concomitantly generated during hydrogen peroxide photosynthesis. The catalysts may show some oxidation on their surface or were damage after continuous photocatalysis, FTIR spectroscopy measurement after photocatalysis should be performed. Moreover, the Co leaching experiments across the reaction time should be included. Also, to highlight the

strength of this work, the authors should also evaluate the stability of pristine Co-CN or reference Co-CN@G and provide a reasonable explanation to clarify the difference in the manuscript.

Response: Thanks for your value comments.

It is true that the aggressive hydroxyl radical and superoxide radical are concomitantly generated during H₂O₂ photosynthesis, which may result in the structural damage of the photocatalysts after continuous photocatalysis. Following the reviewer's suggestion, we have performed the FTIR test of Co-CN@G after 15 continuous cycles, the corresponding spectra was added in the revised manuscript in Supplementary Fig. 25. The recovered sample after repeatedly rinsing displayed the similar spectrum with the fresh sample at room temperature. The absorption peak is not significantly changed, indicating that Co-CN@G exhibited good stability, and oxidative decomposition does not occur obviously. To analyze the Co leaching, Co-CN@G was subjected to the photocatalytic reaction in water for more than 120 h, the Co concentration in the resulting solution was below the detection limitation by ICP-MS measurement, showing the strong anchoring of Co on CN.

Supplementary Figure 25. FTIR spectrum of Co-CN@G before and after photocatalytic cycles.

Additionally, we have evaluated the stability of reference CN@G and pristine

Co-CN with continuous 15 photocatalytic cycles, the corresponding data was added in the revised manuscript in Supplementary Fig. 24. The CN@G exhibits the gradually decreased photocatalytic activity. In sharp contrast, the pristine Co-CN yielded the obviously decreased H₂O₂ amount. These results suggest that the particular integration of Co single atom and CN@G is conducive to the enhanced stability of Co-CN@G during the photocatalytic process.

Supplementary Figure 24. The stability of photocatalysts after continuous photocatalytic reactions: a) CN@G, and b) Co-CN.

4. The authors stated that the excessive amounts of either Co, Co-CN or RGO resulted in an inferior photocatalytic activity, please provide detailed explanation. Co nanoparticle/nanocluster agglomeration could be observed at higher Co loading? Whether the specific surface area and pore distribution significantly affected the catalytic performance? The data need be provided.

Response: Thanks for your value comments.

It was observed that the excessive amounts of Co loading resulted in an inferior photocatalytic activity, which should be attributed to the formation of Co-based nanoclusters and nanoparticles, as revealed by the HADDF-STEM image (Supplementary Fig. 21). As demonstrated, nanoclusters were clearly observed on the samples with 2.8 wt.% Co loading. The formation of nanoclusters or the excessive amounts of Co loading could result in the generation of lots of holes or defect (lower degree of condensation), boosting the recombination of charge carriers. On the other hand, the excessive CN without the intimate interfacial contact with RGO could lead to the insufficient charge separation/migration, which is detrimental to the photocatalytic activity and H₂O₂ yield.

Supplementary Figure 21. HADDF-STEM image of Co-CN@G with 2.8 wt.% Co loading.

The surface area and pore texture of heterogeneous catalysts can significantly affect the catalytic performance. The nitrogen adsorption-desorption isotherms of the as-synthesized Co-CN@G is given in Supplementary Fig. 2. The Co-CN@G possessed a specific surface area of 113.9 m² g⁻¹ and a pore volume of 0.09 cm³ g⁻¹, respectively. It is believed that the higher surface area and pore volume would improve the catalytic performance of catalysts. In our study, the Co-CN@G was synthesized with the use of

porous RGO as the support. The surface area and pore volume of the Co-CN@G were mainly determined by RGO. Thus, the adsorption-desorption behavior of Co-CN@G was difficult to be systematically adjusted by changing the preparation procedures while keeping the optimized ratio of Co, CN and RGO.

Supplementary Figure 2. a) Nitrogen adsorption-desorption isotherms of Co-CN@G, CN@G, and Co-CN. (b) The corresponding pore size distributions.

5. The photocatalysts contained graphene oxide component in their structure. It is well-known that the carbon materials could be aggregated in high salt seawater because of the electric double layer suppression. Such agglomeration has been observed in this work? Please clarify it.

Response: Thanks for your value comments.

It has been reported that nanomaterials are likely to rapidly aggregate in seawater because the high ionic strength of seawater suppresses the electrostatic repulsive forces between particles. The particle agglomeration in seawater or salt water is strongly dependent on the property of the material. We have evaluated the dispersion of the photocatalysts by ultrasonically dispersing them in seawater (50 mg of RGO or Co-CN@G was dispersed into 30 mL of seawater). As observed, the pristine RGO experienced the spontaneous agglomeration in real seawater with the extension of resting time. By contrast, Co-CN@G can be stably dispersed in the real seawater with the prolonged resting time. These results suggest that the coupling of Co-CN and RGO

can prevent the agglomeration of photocatalysts in high salt seawater.

Fig. R4 Diagram of RGO dispersed in seawater at different resting time. a) 0 h, b) 5 h, and c) 10 h.

Fig. R5 Diagram of Co-CN@G dispersed in seawater at different resting time. a) 0 h, and b) 10 h.

- The authors stated that hydrogen peroxide can be produced through oxygen reduction reaction and water oxidation reaction. The latter route was identified by isotopic labeling experiments. In general, water oxidation needs a high potential, theoretically at 1.76 V versus NHE, which has rarely been mentioned in previous studies. It would be important for the reader that the authors should include the energy level of the photocatalysts in the schematic figure, Fig. 5c. Also, the authors should give a brief description on the formation of hydrogen peroxide in the context.

Response: Thanks for your value comments.

It's true that in previous studies, the water oxidation to produce H_2O_2 was rarely mentioned. In our work, it was found that H_2O_2 can be formed through oxygen

reduction reaction and water oxidation reaction, as clearly confirmed by isotopic labeling experiments. It is true that H₂O₂ formed from water oxidation needs a relatively high potential. The valence band energy of Co-CN@G (~2.0 eV) is also sufficient to drive the H₂O₂ production from water oxidation. Following the reviewer's suggestion, the energy level of the photocatalyst has been illuminated in the schematic figure, Fig. 5c. We have added a brief description on the formation pathways of H₂O₂ over Co-CN@G heterostructure in the context marked in red: Upon light illumination, the photoexcited electrons on the CB can drive the H₂O₂ formation from the step-wise two-electron O₂ reduction with •O₂⁻ as the intermediate (O₂/•O₂⁻, -0.33 V). The photogenerated hole from the VB can drive water oxidation via the two-electron and four-electron pathways.

7. The detailed information on the outdoor test should be provided in method section, such as the oxygen partial pressure, solution volume, etc. The figure labels in the inset of Fig. 3i and Fig. 4c are difficult to read. The grammar errors and mistakes in this manuscript need be corrected carefully.

Response: Thanks for your value comments.

We missed the detailed information on the outdoor test in the manuscript. According to the reviewer's suggestion, we have added the detailed information on the outdoor test in the revised manuscript. The figure labels in the inset of Fig. 3i and Fig. 4c are revised. This manuscript was also examined and corrected carefully. We also asked a native English speaker to help us to correct the grammar and word use throughout our manuscript, and the changes made in the manuscript are highlighted in red.

Reviewer #3 (Remarks to the Author):

Wang et al. report on solar H₂O₂ production using C₃N₄ functionalized with Co single atom catalysts. This is an important research question, especially since most photocatalyst get deactivated in seawater. Despite the impressive performance of the catalyst, there several issues that need to be addressed, before the manuscript could be

considered for publication.

Response: Thanks for your inspiring comments and recommendation. Following the reviewer's suggestions, we have carefully and thoroughly revised the manuscript, and the changes made in the manuscript are highlighted in red.

1. The authors present a model for theoretical studies on the catalytic activity with assumed single atom precision (Co). However, the theoretical model lacks the experimental basis, because the authors have determined the nitrogen positions with XPS, which does not tell whether one has N-clustering or not (similar to bipyridine). The N positions determine the bonding of Co and also have an influence on the electronic structure. This can be solely determined according with ^{15}N solid state NMR spectroscopy. Despite the low abundancy of ^{15}N , recent reports have shown that it is relatively easy to obtain solid ^{15}N solid state NMR spectra (from unlabelled N-carbon) if the N content is above ca. 3.5% (Chem. Mater. 2020, 32, 17, 7263-7273, <https://doi.org/10.1021/acs.chemmater.0c01666> and ACS Catal. 2021, 11, 22, 14087-14101, <https://doi.org/10.1021/acscatal.1c03733>). The first paper reports the method and the second even applies it to C_3N_4 photocatalysts for H_2O_2 production, which is as such the ideal reference for the material of the current manuscript. The N position could be also determined for a sample without Co in order to get the correct model for the computational studies. XPS is not the state-of-the-art to determine the type of nitrogen in the material. Atomic precision is required if one wants to set up a model. In addition, the nitrogen species also affect the adsorption of the substrate.

Response: Thanks for your value comments.

It's true that the N positions determine the bonding of Co and also have an influence on the electronic structure and the adsorption of the substrate. In our previous manuscript, XPS has been solely used to determine the type of nitrogen in the material, while it did not tell whether Co-CN@G has N-clustering or not. Given that a single technique usually cannot give comprehensive information, the complementary results are adopted through the use of different tools to better identify the coordination

environment of Co single atom and understand their comprehensive properties.

It has been shown that ^{15}N solid-state nuclear magnetic resonance spectroscopy (^{15}N NMR) is capable of delivering crucial information for the nitrogen inserted in materials (Szewczyk, I. et al. *Chem. Mater.* **32**, 7263-7273 (2020); Zhang, T. et al. *ACS Catal.* **11**, 14087-14101 (2021)). Although this experimental approach is very challenging due to the natural abundance of ^{15}N being only 0.37%, it allows the detection of all types of N sites at the atomic and molecular levels. Following the reviewer's suggestion, we have performed the ^{15}N NMR measurements at natural ^{15}N isotope abundance to gain insights into chemical bonding network of the Co-CN@G. The cross-polarization ^{15}N magic angle spinning NMR spectrum of Co-CN@G is presented in Supplementary Fig. 11. As shown, the spectrum of the Co-CN@G exhibits the poor signal-to-noise ratio, which mainly may be attributed to the interference of conductive RGO and magnetic Co atoms under high-intensity magnetic fields. Alternatively, the ^{15}N NMR spectra of CN (sulfur doped carbon nitride in this study) and Co-CN was collected (Fig. 1h). The spectrum of CN support exhibits four resonance peaks centered at about 107, 135, 156 and 191 parts per million (ppm) chemical shifts. The three characteristic ^{15}N NMR features of CN at 135, 156 and 107 ppm represent the bridged NH, central NC_3 and NH_2 , respectively. The NMR peak at 191 ppm chemical shift could be assigned to NC_2 in the tri-s-triazine ring (Wang, Y. et al. *Adv. Mater.* **33**, 2105482 (2021); Zhang, P. et al. *Angew. Chem.* **132**, 16343-16351 (2020)). Anchoring Co onto the CN support result in a substantial decrease in the number of NC_2 sites, as evidenced by the large drop in the intensity of the 191-ppm NMR peak. These results strongly suggest that Co atoms bond to the NC_2 sites on the CN surface through Co-N bridges. Furthermore, it reveals that there is no N-clustering similar with bipyridine in Co-CN@G. However, the coordination environment needs to be further clarified because of the presence of sulfur doping within CN network.

Supplementary Figure 11. ^{15}N solid-state NMR MAS spectra of Co-CN@G.

Fig. 1h ^{15}N solid-state NMR MAS spectra of Co-CN and pristine CN.

X-ray absorption near-edge structure (XANES) and extended X-ray absorption fine structure (EXAFS) measurement were performed at Co *K*-edge to further investigate the local structures of Co atoms. EXAFS of Co-CN@G exhibited a dominant peak at $\sim 1.82 \text{ \AA}$, which can be separated into two independent peaks assigned to Co-N and Co-S scattering. The strong Co-N and Co-S coordination can be observed in the WT contour plots. The FT-EXAFS profile (Fig. 1g) and the corresponding fitting results (Supplementary Table 1) verified that CoN_1S_3 (one Co-N bond and three Co-S bonds) should be the dominating structure in Co-CN@G. The Co-N and Co-S distance

was well-fitted to be 1.94 and 2.29 Å, respectively.

The model for DFT theoretical studies in this manuscript was established based on the best-fitted coordination number of XAFS characterizations, which has been typically employed in many previous studies. The model of Co-CN@G used for DFT theoretical studies is based on the single Co atom coordinated to one neighboring N atom and three S atoms. We choose the multiple layer C₃N₄ structure as the substrate to mimic the structures of Co single atoms on C₃N₄, because it is more stable than the single layer one, which has been fully discussed in the previous studies (Teng, Z. et al. *Nat. Catal.* **4**, 374-384 (2021); Zhang, P. et al. *Nat. Commun.* **10**, 940 (2019)). The Co atom locates at the hole site of sulfur doped C₃N₄. The catalyst structure is optimized according to the literature (Cao, L. et al. *Nat. Commun.* **10**, 4849 (2019); Yang, T. et al. *Nat. Commun.* **12**, 6022 (2021); Tian, S. et al. *J. Am. Chem. Soc.* **140**, 11161-11164 (2018)). After structural relaxation, Co atom strongly coordinates with one N atom with the bond length of 1.96 Å, and three longer Co-S bonds with the average bond length of 2.28 Å, which are very close to the fitting results from EXAFS measurement. In particular, the optimized results indicate that the Co-N originates from the coordination between Co and NC₂ within CN, and such coordination is the most stable structure after relaxation. This was also consistent with the ¹⁵N NMR result in this study and is similar with the previous works of the pristine carbon nitride supported metal-single atom (Tian, S. et al. *Nat. Commun.* **12**, 3181 (2021); Liu, H. et al. *Nano Lett.* **21**, 10284-10291 (2021); Chen, P. et al. *ACS Nano* **14**, 15841-15852 (2020)). Overall, thanks again for the reviewer's good recommendation.

2. Fig 1b shows the TEM EDX mapping, but there is no information on the chemical composition. What is the C/N ratio?

Response: Thanks for your value comments.

The important information on the chemical composition of Co-CN@G has been missed in our previous manuscript. Following the reviewer's suggestion, we have added it in this revised manuscript depicted in Supplementary Fig. 6. As shown, the C/N atomic ratio of Co-CN@G was 2.75.

3. Supplementary table S1 is not mentioned in the manuscript and it states the current catalyst to have the highest activity. One cannot really compare a catalyst that is based on heavier elements with a catalyst that is mainly composed of carbon and nitrogen. This should be rather related to activity/mole. I understand that this is quite difficult, but a sentence should be mentioned here.

Response: Thanks for your value comments.

As demonstrated in Supplementary Table 2, Co-CN@G yielded significantly higher solar-to-chemical efficiency than other particulate photocatalysts in pure water and natural seawater. It is true that one cannot really compare a catalyst that is based on heavier elements with a catalyst that is mainly composed of carbon and nitrogen. In general, a catalyst's specific/mole activity can be useful when attempting to compare the intrinsic activity of catalysts with different surface areas or loadings. However, the as-reported methods employed to evaluate H₂O₂-evolving catalysts are not standardized due to the various definitions of specific activity, making it difficult to compare the activity of these materials. Follow the reviewer's suggestion, a sentence on the activity statement has been mentioned in the manuscript.

4. The ¹⁸O₂ experiments still show lots of ¹⁶O₂ and the authors claim that this stems from oxygen which is present in air during sample preparation. I agree that the ¹⁸O₂ tests are a proof, but I would suggest to make one additional test as with a phosphine capping agent and then to detect the m/z value.

Answer: Thank you for your valuable suggestion.

Follow the reviewer's suggestion, we have performed the additional experiments for ¹⁸O₂ labeling tests using triphenylphosphine (PPh₃) as the capping agent to examine the oxygen source of H₂O₂. In fact, H₂O₂ is known to react with PPh₃ to give triphenylphosphine oxide (OPPh₃). The PPh₃ sample co-incubated with the formed H₂O₂ solution from the labeled ¹⁸O₂ was analyzed by gas chromatography coupled to mass spectrometry (GCMS). The corresponding m/z result for ¹⁸O₂ labelled reaction is displayed in Supplementary Fig. 32.

As demonstrated, both peaks for $^{16}\text{O}=\text{PPh}_3$ and $^{18}\text{O}=\text{PPh}_3$ were observed from the GCMS spectrum (Supplementary Fig. 32) when the H_2O_2 generated from the labeled $^{18}\text{O}_2$ was made to react with PPh_3 , and the relative abundance of $^{18}\text{O}=\text{PPh}_3$ is significantly higher than that of $^{16}\text{O}=\text{PPh}_3$. The presence of intensified $^{18}\text{O}=\text{PPh}_3$ proves that the main oxygen source of H_2O_2 was the O_2 gas. The concomitant formation of $^{16}\text{O}=\text{PPh}_3$ reveals that H_2O_2 can be generated from the water oxidation (the direct water oxidation into H_2O_2 ; O_2 generated from water further participated in the formation of H_2O_2).

Supplementary Figure 32. GC-MS spectrum of the $^{18}\text{O}_2$ labelled reaction using triphenylphosphine as the capping agent.

5. The authors write: “In summary, we reported a photothermal-photocatalytic Co-CN@G particulate photocatalyst for highly efficient non-sacrificial H_2O_2 production from natural seawater. By taking advantage of the cooperation of CN@G heterostructure and isolated Co single atoms, this system exhibited excellent and stable catalytic activity towards H_2O_2 photosynthesis”. If you take a look at Fig. 3b, then you see that the catalyst gets deactivated after 6 h significantly. I am not that convinced that one can speak here about a stable catalyst in seawater; this resembles rather the previously published catalyst. I would recommend to rephrase the claim of the manuscript. Otherwise, it is a bit misleading regarding the stability.

Answer: Thank you for your good comment and suggestion.

In our study, it was observed from Fig. 3b that H₂O₂ production on the optimized Co-CN@G dramatically increased with time at the early stage of irradiation, and the change was not very obvious after 6 h reaction. This result should be attributed to the production-decomposition balance because H₂O₂ will experience the decomposition reaction into the O₂ and water, as revealed by the decomposition curve in Supplementary Fig. 22. In addition, the recovered Co-CN@G was stable enough to maintain its photocatalytic activity during 15 repeated cycles. However, it is still premature to claim that Co-CN@G is a stable catalyst in seawater for photocatalytic H₂O₂ production because of the relative short reaction time compared to other catalysts in the previous reports. Follow the reviewer's suggestion, we have re-phrased the claim of the manuscript on the stability.

REVIEWERS' COMMENTS

Reviewer #2 (Remarks to the Author):

The authors have fully revised the manuscript according to the reviewer's comments. I found they have added some necessary experiments, characterizations and references to answer our concerns such as catalytic centers, temperature dependence, stability, specific surface surface area, labeling test for oxygen source, chemical somposition, and so on. The conclusions are fully supported. And now this manuscript can be considered for publication in Nat. Commun.

Reviewer #3 (Remarks to the Author):

The authors have carried out additional experiments to strengthen the claims and sufficiently addressed the issues that I pointed out in the previous review. I therefore would recommend the manuscript for publication in its current form.

Response to Reviewers' Comments

Reviewer #2:

The authors have fully revised the manuscript according to the reviewer's comments. I found they have added some necessary experiments, characterizations and references to answer our concerns such as catalytic centers, temperature dependence, stability, specific surface area, labeling test for oxygen source, chemical composition, and so on. The conclusions are fully supported. And now this manuscript can be considered for publication in Nat. Commun.

Response: We thank the reviewer for taking time to review our manuscript and the positive comments on the manuscript.

Reviewer #3:

The authors have carried out additional experiments to strengthen the claims and sufficiently addressed the issues that I pointed out in the previous review. I therefore would recommend the manuscript for publication in its current form.

Response: We thank the reviewer for taking time to review our manuscript and the positive comments on the manuscript.